# ON POSITION EMBEDDINGS IN BERT

**Benyou Wang**
University of Padova
wang@dei.unipd.it

**Lifeng Shang**
Huawei Noah's Ark Lab
Shang.Lifeng@huawei.com

**Christina Lioma**
University of Copenhagen
c.lioma@di.ku.dk

**Xin Jiang**
Huawei Noah's Ark Lab
Jiang.Xin@huawei.com

**Hao Yang**
Huawei Technologies Co., Ltd.
yanghao30@huawei.com

**Qun Liu**
Huawei Noah's Ark Lab
qun.liu@huawei.com

**Jakob Grue Simonsen**
University of Copenhagen
simonsen@di.ku.dk

## ABSTRACT

Various Position Embeddings (PEs) have been proposed in Transformer based architectures (e.g. BERT) to model word order. These are empirically-driven and perform well, but no formal framework exists to systematically study them. To address this, we present three properties of PEs that capture word distance in vector space: *translation invariance*, *monotonicity*, and *symmetry*. These properties formally capture the behaviour of PEs and allow us to reinterpret sinusoidal PEs in a principled way. Moreover, we propose a new probing test (called 'identical word probing') and mathematical indicators to quantitatively detect the general attention patterns with respect to the above properties. An empirical evaluation of seven PEs (and their combinations) for classification (GLUE) and span prediction (SQuAD) shows that: (1) both classification and span prediction benefit from translation invariance and local monotonicity, while symmetry slightly decreases performance; (2) The fully-learnable absolute PE performs better in classification, while relative PEs perform better in span prediction. We contribute the first formal and quantitative analysis of desiderata for PEs, and a principled discussion about their correlation to the performance of typical downstream tasks.

## 1 INTRODUCTION

Position embeddings (PEs) are crucial in Transformer-based architectures for capturing word order; without them, the representation is bag-of-words. Fully learnable absolute position embeddings (APEs) were first proposed by Gehring et al. (2017) to capture word position in Convolutional Seq2seq architectures. Sinusoidal functions were also used with Transformers to parameterize PEs in a fixed ad hoc way (Vaswani et al., 2017). Recently, Shaw et al. (2018) used relative position embedding (RPEs) with Transformers for machine translation. More recently, in Transformer pre-trained language models, BERT (Devlin et al., 2018; Liu et al., 2019) and GPT (Radford et al., 2018) used fully learnable PEs. Yang et al. (2019) modified RPEs and used them in the XLNet pre-trained language model. To our knowledge, the fundamental differences between the various PEs have not been studied in a principled way.

We posit that the aim of PEs is to capture the sequential nature of positions in vector space, or technically, to bridge the distances in $\mathbb{N}$ (for positions) and $\mathbb{R}^D$ (for position vectors). We therefore propose three expected properties for PEs: *monotonicity*, *translation invariance*, and *symmetry* [1]. Using these properties, we formally reinterpret existing PEs and show the limitations of sinusoidal

---

[1]Informally, as positions are originally positive integers, one may expect position vectors in vector space to have the following properties: 1) neighboring positions are embedded closer than faraway ones; 2) distances of two arbitrary $m$-offset position vectors are identical; 3) the metric (distance) itself is symmetric.

PEs (Vaswani et al., 2017): they cannot adaptively meet the *monotonicity* property – thus we propose learnable sinusoidal PEs.

We benchmark 13 PEs (including APEs, RPEs, and their combinations) in GLUE and SQuAD, in a total of 11 individual tasks. Several indicators are devised to quantitatively measure translation invariance, monotonicity, and symmetry, which can be further used to calculate their statistical correlations with empirical performance in downstream tasks. We empirically find that both text classification tasks (in GLUE) and span prediction tasks (SQuAD V1.0 and V 2.0) can benefit from monotonicity (in nearby offset) and translation invariance (in particular without considering special tokens like `[CLS]`), but symmetry decreases performance since it can not deal with directions between query vectors and key vectors when calculating attentions. Plus, models with unbalanced attention regarding directions (generally attending more to *preceding* tokens than to *succeeding* tokens) slightly correlate with better performance (especially for span prediction tasks).

Experiments also show that the *fully-learnable APE performs better in classification, while RPEs perform better in span prediction tasks*. This is explained by our proposed properties as follows: RPEs perform better in span prediction tasks since they meet better translation invariance, monotonicity , and asymmetry; the fully-learnable APE which does not strictly have the *translation invariance* and monotonicity properties during parameterizations (as it also performed worse in measuring translation invariance and local monotonicity than other APEs and all RPEs) still performs well because it can flexibly deal with special tokens (especially, unshiftable `[CLS]`).

Regarding the newly-proposed learnable sinusoidal PEs, the learnable sinusoidal APE satisfies the three properties to a greater extent than other APE variants, and the learnable sinusoidal RPE exhibits better direction awareness than other PE variants. Experiments show that BERT with sinusoidal APEs slightly outperforms the fully-learnable APE in span prediction, but underperforms in classification tasks. Both for APEs and RPEs, learning frequencies in sinusoidal PEs appears to be beneficial. Lastly, sinusoidal PEs can be generalized to treat longer documents because they completely satisfy the translation invariance property, while the fully-learnable APE does not.

The contributions of this paper are summarised below: 1) We propose three principled properties for PEs that are either formally examined or empirically evaluated by quantitative indicators in a novel Identical Word Probing test; 2) We benchmark 13 PEs (including APEs, RPEs and their combinations) in GLUE, SQuAD V1.1 and SQuAD V2.0, in a total of 11 individual tasks; 3) we experimentally evaluate how the performance in individual tasks benefits from the above properties; 4) We propose two new PEs to extend sinusoidal PEs to learnable versions for APEs/RPEs.

## 2 PROPERTIES OF POSITION EMBEDDINGS

Gehring et al. (2017); Vaswani et al. (2017) use absolute word positions as additional features in neural networks. Positions $x \in \mathbb{N}$ are distributively represented as an *embedding* of $x$ as an element $\vec{x} \in \mathbb{R}^D$ in some Euclidean space. By standard methods in representation learning, similarity between embedded objects $\vec{x}$ and $\vec{y}$ is typically expressed by an inner product $\langle \vec{x}, \vec{y} \rangle$, for instance the dot product gives rise to the usual cosine similarity between $\vec{x}$ and $\vec{y}$. Generally, if words appear close to each other in a text (i.e., their positions are nearby), they are more likely to determine the (local) semantics together, than if they occurred far apart. Hence, *positional proximity* of words $x$ and $y$ should result in proximity of their embedded representations $\vec{x}$ and $\vec{y}$. One common way of formalizing this is that an embedding should preserve the *order* of distances among positions [2]. We denote $\phi(\cdot, \cdot)$ as a function to calculate closeness/proximity between embedded positions, and any inner product can be a special case of $\phi(\cdot, \cdot)$ with good properties. We can express preservation of the order of distances as: For every $x, y, z \in \mathbb{N}$,

$$|x - y| > |x - z| \implies \phi(\vec{x}, \vec{y}) < \phi(\vec{x}, \vec{z}) \tag{1}$$

Note that on the underlying space, the property in Eq. (1) has been studied for almost 60 years (Shepard, 1962), in both algorithmics (Bilu & Linial, 2005; Badoiu et al., 2008; Maehara, 2013), and machine learning (Terada & Luxburg, 2014; Jain et al., 2016) under the name *ordinal embedding*. As we are interested in the simple case of positions from $\mathbb{N}$, Eq. (1) reduces to the following property:

---

[2]Theoretical evidence for this is nontrivial unless we assume more about the particular non-linear functions. We empirically find that all learned PEs can preserve the order of distance

**Property 1.** **Monotonicity**: The proximity of embedded positions decreases when positions are further apart:

$$\forall x, m, n \in \mathbb{N} : m > n \iff \phi(\vec{x}, \overrightarrow{x+m}) < \phi(\vec{x}, \overrightarrow{x+n}) \tag{2}$$

A priori, a position embedding might treat every element $\mathbb{N}$ individually. However, considering pairs of positions based on their *relative* proximity (rather than the absolute value of the positions), can lead to simplified and efficient position embeddings (Wang et al., 2020). Such embeddings satisfy *translation invariance*:

**Property 2.** **Translation invariance**: The proximity of embedded positions are translation invariant:

$$\forall x_1, \ldots, x_n, m \in \mathbb{N} : \phi(\vec{x}_1, \overrightarrow{x_1+m}) = \phi(\vec{x}_2, \overrightarrow{x_2+m}) = \cdots = \phi(\vec{x}_n, \overrightarrow{x_n+m}) \tag{3}$$

Finally, since the inner product is symmetric, we also consider whether $\phi(\cdot, \cdot)$ is symmetric:

**Property 3.** **Symmetry**: The proximity of embedded positions is symmetric,

$$\forall x, y \in \mathbb{N} : \phi(\vec{x}, \vec{y}) = \phi(\vec{y}, \vec{x}) \tag{4}$$

There is no generally accepted standard set of properties for position embeddings; based on prior work as described above, we posit that the above properties are important, and now examine several existing PEs in relation to these properties, either formally (in Sec. 3) or empirically (in Sec. 4).

## 3 UNDERSTANDING PES VIA THE PROPERTIES

PEs come in two variants: *absolute* PEs (APEs) where single positions are mapped to elements of the representation space, and *relative* PEs (RPEs) where the *difference* between positions (i.e., $x - y$ for $x, y \in \mathbb{N}$) is mapped to elements of the embedding space. For Transformer-based architectures, the difference between APEs and RPEs manifests itself in the attention mechanism, in particular how the matrices of query, key, and value weights $W^Q$, $W^K$, and $W^V$ are used to calculate attention in each attention head. Consider two positions $x, y \in \mathbb{N}$, let $\text{WE}_x$ be the word embedding of the word at position $x$, and let $P_x$ and $P_{x-y}$ be the embeddings of the position $x$ and relative position $x - y$, respectively. The query-key-value vector for the word at position $x$ is typically calculated as below for APEs and RPEs[3] respectively:

$$\text{APE: } \begin{bmatrix} Q_x \\ K_x \\ V_x \end{bmatrix} = (\text{WE}_x + P_x) \odot \begin{bmatrix} W^Q \\ W^K \\ W^V \end{bmatrix} \quad ; \quad \text{RPE: } \begin{bmatrix} Q_x \\ K_x \\ V_x \end{bmatrix} = \text{WE}_x \odot \begin{bmatrix} W^Q \\ W^K \\ W^V \end{bmatrix} + \begin{bmatrix} \mathbf{0} \\ P_{x-y} \\ P_{x-y} \end{bmatrix} \tag{5}$$

Observe that while the APEs calculation is linear in $(W^Q, W^K, W^V)$ with the word and position embeddings merged into the coefficient, the RPEs calculation is affine, with the relative position embedding $P_{x-y}$ acting as an offset independent of the word embedding $\text{WE}_x$.

In Transformers, the resulting representation is a sum of value vectors with weights depending on $A = QK^T$, that is, $\text{Attention}(Q, K, V) = \text{softmax}(QK^T / \sqrt{d_k})V$. In the rest of the paper, we examine PEs in the above architecture with respect to the properties introduced in Section 2. In particular, we study four well-known variants of PEs: (1) the **fully learnable APE** (Gehring et al., 2017), (2) the **fixed sinusoidal APE** (Vaswani et al., 2017), (3) the **fully learnable RPE** (Shaw et al., 2018), and (4) the **fixed sinusoidal RPE** (Wei et al., 2019).

### 3.1 UNDERSTANDING SINUSOIDAL PES

With a sinusoidal parameterization in PEs, we may use a specific proximity, i.e., an efficient inner product like a dot product, to check if the sinusoidal form of PEs meets the above properties. The dot product between any two position vectors is

$$A_{x,y} = \langle \vec{x}, \vec{y} \rangle = \text{sum}\left(\begin{bmatrix} \sin(\omega_1 x) \\ \cos(\omega_1 x) \\ \cdots \\ \sin(\omega_{\frac{D}{2}} x) \\ \cos(\omega_{\frac{D}{2}} x) \end{bmatrix} \odot \begin{bmatrix} \sin(\omega_1 y) \\ \cos(\omega_1 y) \\ \cdots \\ \sin(\omega_{\frac{D}{2}} y) \\ \cos(\omega_{\frac{D}{2}} y) \end{bmatrix}\right) = \text{sum}\left(\begin{bmatrix} \sin(\omega_1 x)\sin(\omega_1 y) \\ \cos(\omega_1 x)\cos(\omega_1 y) \\ \cdots \\ \sin(\omega_{\frac{D}{2}} x)\sin(\omega_{\frac{D}{2}} y) \\ \cos(\omega_{\frac{D}{2}} x)\cos(\omega_{\frac{D}{2}} y) \end{bmatrix}\right) = \sum_{i=0}^{\frac{D}{2}} \cos(\omega_i(x-y)) \tag{6}$$

---

[3]There are many variants of RPEs (e.g., (Dai et al., 2019)). As selecting RPEs is not the main concern in this paper, we give the original (and typical) RPEs only. One can easily extend this work to other RPE variants.

Table 1: Overview of PEs. $P_x$ or $P(x)$ is the $x$-th absolute/relative position vector (the latter is parameterized by sinusoidal functions). The newly-proposed PEs in this paper are in bold.

| PEs | formulation | parameter scale |
|---|---|---|
| fully learnable APE (Gehring et al., 2017) | $P_x \in \mathbb{R}^D$ | $L \times D$ |
| fixed sinusoidal APE (Vaswani et al., 2017) | $P(x) = [\cdots, \sin(\omega_i x), \cos(\omega_i x), \cdots]^T;$ $\omega_i = (1/10000)^{2i/D}$ | 0 |
| **learnable sinusoidal APE** | $P(x) = [\cdots, \sin(\omega_i x), \cos(\omega_i x) \cdots]^T;$ $\omega_i \in \mathbb{R}$ | $\frac{D}{2}$ |
| fully learnable RPE (Shaw et al., 2018) | $P_x \in \mathbb{R}^D$ | $L \times D$ |
| fixed sinusoidal RPE (Wei et al., 2019) | $P(x) = [\cdots, \sin(\omega_i x), \cos(\omega_i x), \cdots]^T;$ $\omega_i = (1/10000)^{2i/D}$ | 0 |
| **learnable sinusoidal RPE** | $P(x) = [\cdots, \sin(\omega_i x), \cos(\omega_i x), \cdots]^T;$ $\omega_i \in \mathbb{R}$ | L |

Note that sinusoidal PEs satisfy both Property 2 (*translation invariance*) because the inner product is only associated with its position difference $x - y$, and Property 3 (*symmetry*), because the dot product itself is symmetric: $\langle \vec{x}, \vec{y} \rangle = \langle \vec{y}, \vec{x} \rangle$. Note also that checking Property 1 is equivalent to checking monotonicity of the map $\psi(m) = \sum_{i=1}^{D/2} \cos(\omega_i m)$. $\psi(m)$ is monotone on intervals where its first order derivative $\psi'(m) = \sum_{i=1}^{D/2} -\omega_i \sin(\omega_i m)$ does not change sign, and these intervals depend on the choice of $\omega_i$. With fixed frequencies $\omega_i = (1/10000)^{2i/D}$, it is monotonous when $m$ is roughly between 0 and 50, indicating that it can only strictly perceive a maximum distance of 50 and it is insensitive to faraway distances (e.g. longer than 50).

Although sinusoidal PEs with fixed frequencies (i.e., $\omega_i = (1/10000)^{2i/D}$) are common in APEs and RPEs, we argue that learning these frequencies is useful because it can adaptively adjust intervals of monotonicity (they do not have to be 0-50 as in the fixed sinusoidal APE) [4]. With trainable frequencies, we can adaptively allocate a number of frequencies in a data-driven way. App. A.2 explains the expressive power of sinusoidal PEs with trainable frequencies from the perspective of the Fourier series. Extending existing fixed sinusoidal PEs to a learnable version with learnable frequencies gives two variants: a **learnable sinusoidal APE** and a **learnable sinusoidal RPE**.

## 3.2 UNDERSTANDING RPEs

RPEs ignore the absolute position of words and directly encode their relative distance. The RPEs expression adheres to the *translation invariance* property during parameterization, since relative distance with the same offset will be embedded as the same embedding, namely, $P_{x_1-y_1} = P_{x_2-y_2}$ if $x_1 - y_1 = x_2 - y_2$. Plus, RPEs that separately embed forward and backward relative embeddings, i.e., $P_{i-j} \neq P_{j-i}$, do not meet symmetry during parameterization.

Sinusoidal RPEs can also embed neighboring relative position in close vectors with a local *monotonicity*, similarly to sinusoidal APEs. Note that the dot products between two sinusoidal relative position vectors with the same offset, without distinguishing positive negative relative position vectors, should be identical [5]. This makes it hardly perceive of the border between preceding and succeeding relative position vectors.

## 4 EXAMINING PE PROPERTIES IN PRE-TRAINED LANGUAGE MODEL

We train BERT with six basic PEs as in Tab. 1 and their combination variants, and conduct a probing test to check to which degree they satisfy the properties.

**Pre-training** The pre-trained "BERT-base-uncased" checkpoint (Devlin et al., 2018) is used to train by replacing the original absolute PE module with a new PE variant (including APEs and RPEs). We train the new models with a sequence length of 128 for 5 epochs and then 512 for another 2 epochs. The training is the same as in the original BERT, i.e., BooksCorpus and Wikipedia (16G raw documents) with whole word masking. To be fair, the BERT with the original fully-learnable

---

[4]See App. A to intuitively understand specific functions of each frequency $\omega_i$

[5]Namely, $\langle P_{x_1-y_1}, P_{x_2-y_2} \rangle = \langle P_{x_3-y_3}, P_{x_4-y_4} \rangle$ if $(x_1 - y_1) - (x_2 - y_2) = (x_3 - y_3) - (x_4 - y_4) = m$, in both $x - y > 0$ and $x - y < 0$.

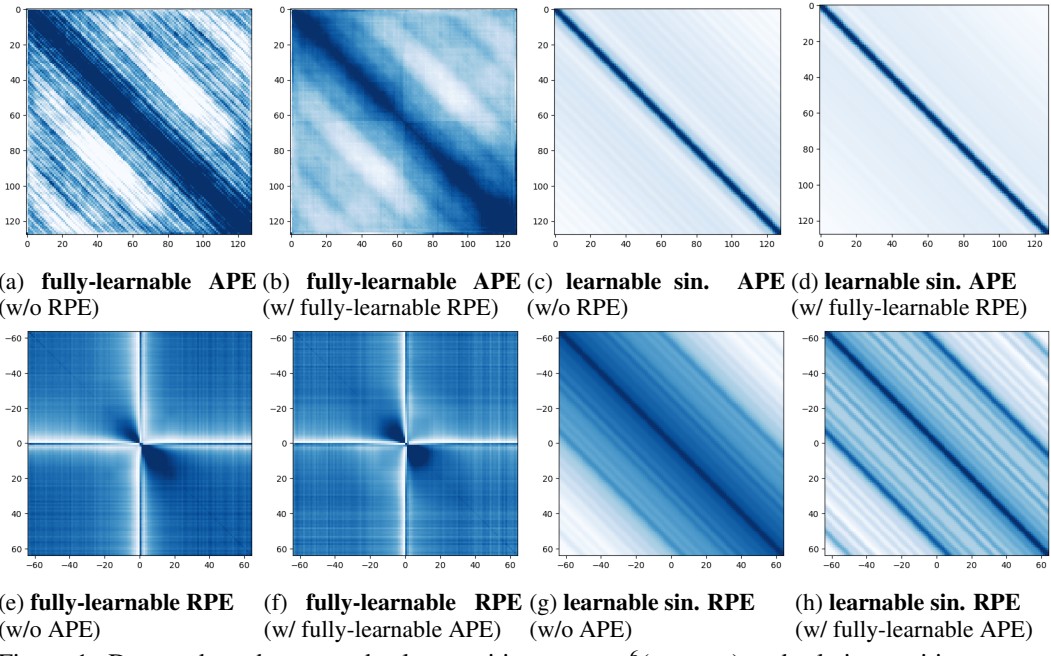

(a) **fully-learnable APE** (w/o RPE)
(b) **fully-learnable APE** (w/ fully-learnable RPE)
(c) **learnable sin. APE** (w/o RPE)
(d) **learnable sin. APE** (w/ fully-learnable RPE)

(e) **fully-learnable RPE** (w/o APE)
(f) **fully-learnable RPE** (w/ fully-learnable APE)
(g) **learnable sin. RPE** (w/o APE)
(h) **learnable sin. RPE** (w/ fully-learnable APE)

Figure 1: Dot products between absolute position vectors [6](top row) and relative position vectors (bottom row). Darker means the two position vectors are closer.

APE is also further trained in the same way. All models have about 110M parameters corresponding to a typical *base* setting, with minor differences solely depending on the parameterization in Tab. 1.

## 4.1 DOT PRODUCT BETWEEN POSITION VECTORS

**APEs** We calculate dot products between two arbitrary position vectors for APEs and RPEs (see Fig. 1). For APEs, neighboring position vectors are generally closer compared to faraway ones. This trend is clearer in the *learnable sinusoidal APE*, which imposes a strict sinusoidal regularization for PEs. Note that additionally adopting RPEs does not affect too much PE patterns, as can be seen by comparing Fig. 1(a) and 1(b), or Fig. 1(c) and 1(d).

**RPEs** In the *fully-learnable RPE* setting, the vertical and horizontal bright bands in 1(e) and 1(f) show that the relative position vectors for small offsets (e.g., $\{P_{-5}, \cdots, P_0, \cdots P_5\}$ ) are notably different to other relative position vectors; it indicates that the relative position vectors with small offsets are more distinguishable than faraway relative position vectors. The four dark corners in 1(e) and 1(f) means that relative position vectors with longer offset than 20, i.e., from -64 to -20 and from 20 to 64, are very close, showing that the *fully-learnable RPE* does not significantly distinguish far-distant RPEs. This suggests that truncating RPEs into a fixed distance (e.g. 64 in (Shaw et al., 2018)), is reasonable. This effect is further explained in App. D.

## 4.2 IDENTICAL WORD PROBING

In APEs, the attention matrix ($A = \text{softmax}(QK^T)$) is related to individual words and their positions, an element of (inactivated) $A$ in the first layer is given by:

$$
\begin{aligned}
a_{ij} &= (w_i + p_i)W^{Q,1}((w_j + p_j)W^{K,1})^T \\
&= \underbrace{w_i W^{Q,1}(W^{K,1})^T w_j^T}_{\text{word-word correspondence}} + \underbrace{w_i W^{Q,1}(W^{K,1})^T p_j^T}_{\text{word-position correspondence}} + \underbrace{p_i W^{Q,1}(W^{K,1})^T w_j^T}_{\text{word-position correspondence}} + \underbrace{p_i W^{Q,1}(W^{K,1})^T p_j^T}_{\text{position-position correspondence}} \quad (7)
\end{aligned}
$$

**Identical word probing for PEs** To study the effect of only PEs in $A$ without considering individual words, we use *identical word probing*: feed many repeated identical words (can be arbitrary,

---

[6]In all figures we only show the first 128 positions instead of 512 positions since they are in principle compatible. Practically in BERT, there is a minor discrepancy between the first 128 positions and the remaining positions due to the typical training strategy (first training on 128-length input and then 512-length input).

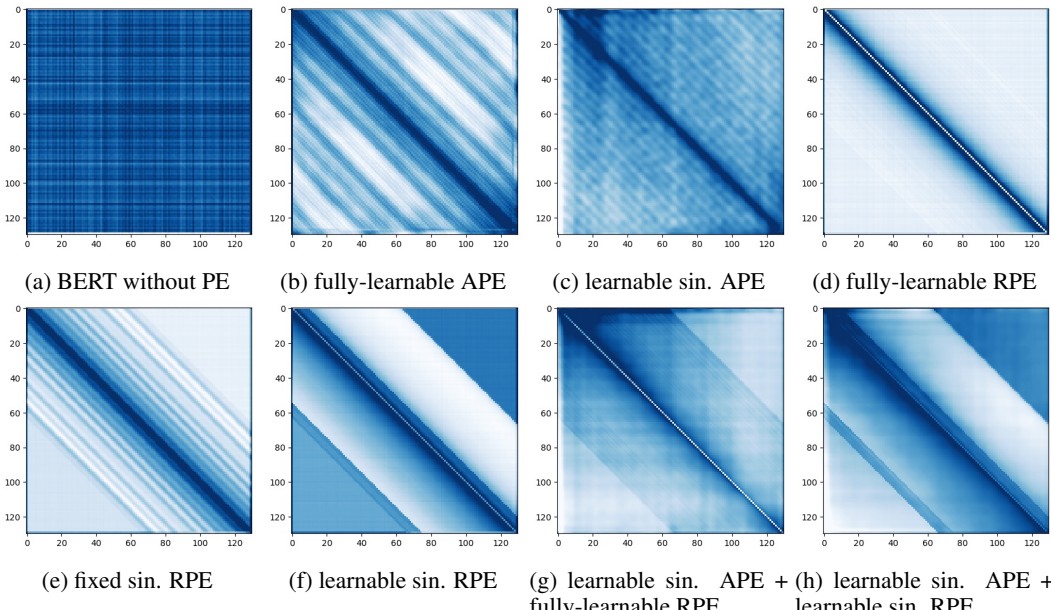

| (a) BERT without PE | (b) fully-learnable APE | (c) learnable sin. APE | (d) fully-learnable RPE |

| (e) fixed sin. RPE | (f) learnable sin. RPE | (g) learnable sin. APE + fully-learnable RPE | (h) learnable sin. APE + learnable sin. RPE |

Figure 2: Identical word probing. Darker in the $i$-th row and $j$-th column means that the $i$-th words generally attend more on the $j$-th words.

denoted as $\bar{w}$) as a sentence to BERT to check the attention values $\bar{A}^{(1)}$, with each element

$$\bar{a}_{ij}^1(\bar{w}) = (\bar{w} + p_i)W^{Q,1}((\bar{w} + p_j)W^{K,1})^T \qquad (8)$$

As we take an average of $\bar{A}^{(1)}$ over many randomly-selected words $\bar{w}$, the general patterns of $\bar{A}^{(1)}$ will not be affected by any particular word. Namely, $\bar{A}^{(1)}$ is word-free and only related to learned PEs. Thus, $\bar{A}^{(1)}$ can be treated as a general attention bias and can also implicitly convey position-wise proximity in Transformers. Note that the probing test could also be applied to RPEs.

### 4.2.1 QUALITATIVE ANALYSIS

Fig. 2 shows the average attention weights among all heads in the first layer. *BERT without PE* nearly treats all words uniformly (bag-of-words). Almost all APEs and RPEs have a clear pattern of translation invariance, local monotonicity in a neighboring window, and symmetry. Note that this is nontrivial since no specific constraints or priors were imposed on fully-learnable APEs/RPEs [7].

BERT with APEs does not show any direction awareness since Fig. 2(b) and 2(c) are nearly symmetrical. As seen from Fig. 2(f,h), BERT with *learnable sinusoidal RPE* generally attends more on forward tokens than backward tokens, which cannot be clearly found in *fully-learnable RPE* and *fixed sinusoidal RPE*. Interestingly, the white bands along the diagonal in Fig. 2 (d, f, g) suggest that some words generally do not attend to themselves, as previously observed in (Clark et al., 2019) [8].

### 4.2.2 QUANTITATIVE ANALYSIS

Using the activated attention values $\bar{A}^{(1)}$ in Eq. 8 [9] we adopt three quantitative indicators to measure to which extent BERT models with individual PEs satisfy the three properties and their derivative indicators (see App. B for details of calculating these indicators) in Tab. 2. Basically, all APEs and RPEs satisfy monotonicity in small offsets and translation invariance compared to *BERT without PE*; All PEs nearly satisfy symmetry except for the learnable sinusoidal RPE and its combinations.

**APEs and RPEs** The learnable sinusoidal APE better satisfies all three properties than fully-learnable APE and fixed sinusoidal APE; this is due to its sinusoidal parameterization and flexible frequencies. RPEs satisfy translation invariance to a higher degree than APEs, because they directly

---

[7]See App. G for an example of the evolution of PE patterns in fully learnable APE which starts from random initialization and ends with some patterns reflecting the properties.

[8]See App. J for more details about the white band effect along the diagonal.

[9]300 random-selected words were used to calculate average $\bar{A}^{(1)}$, as we empirically found that adopting more words almost does not change $\bar{A}^{(1)}$.

Table 2: Quantitative measurement of the properties (monotonicity, translation invariance, symmetry, and direction balance [10]). For these property indicators, the smaller the number, the better the property is met. 0 denotes that the property is ideally satisfied. Direction balance denotes the ratio between the sum of attention values for forward attending and backward attending. 1 means it is fully-balanced in directions. We have indicated the numbers that most closely correspond to satisfaction properties and direction balance for each group in **bold**.

| PEs | monotonicity | | translation invariance | | symmetry | direction balance |
|---|---|---|---|---|---|---|
| | all offsets | first 20 offsets | w/ [CLS] | w/o [CLS] | | |
| BERT without PE | 0.5430 | 0.1393 | 0.9497 | 0.9939 | 0.0005 | 1.0136 |
| BERT-style APE | 0.2461 | 0.0208 | 0.5030 | **0.0143** | 0.0012 | 1.1940 |
| fixed sin. APE | 0.1937 | **0.0190** | 0.2552 | 0.2143 | 0.0010 | **1.0266** |
| learnable sin. APE | **0.1936** | 0.0237 | **0.0653** | 0.0378 | **0.0004** | 1.0281 |
| fully-learnable RPE | 0.1576 | **0.0048** | 0.1178 | **0.0007** | 0.0007 | 1.1930 |
| fixed sin. RPE | **0.1273** | 0.0054 | **0.0924** | 0.0020 | **0.0007** | **1.1565** |
| learnable sin. RPE | 0.3157 | 0.0057 | 0.1397 | 0.0038 | 0.0014 | 1.3223 |
| BERT-style APE + fully-learnable RPE | 0.1993 | **0.0071** | 0.2601 | **0.0059** | 0.0009 | 1.1971 |
| BERT-style APE + fixed sin. RPE | **0.1579** | 0.0143 | **0.1376** | 0.0072 | **0.0007** | **1.1302** |
| BERT-style APE+ learnable sin. RPE | 0.2364 | 0.0158 | 0.2334 | 0.0088 | 0.0014 | 1.3804 |
| learnable sin. APE + fully-learnable RPE | 0.1248 | 0.0065 | 0.0487 | 0.0238 | 0.0007 | 1.1196 |
| learnable sin. APE + fixed sin. RPE | **0.0746** | **0.0040** | **0.0243** | 0.0168 | **0.0007** | **1.0773** |
| learnable sin. APE + learnable sin. RPE | 0.1796 | 0.0052 | 0.0399 | 0.0252 | 0.0027 | 1.6722 |

satisfy translation invariance during parameterization. In the last column, direction balance values of all PEs except for the fixed sin. APE are larger than one, which indicates that BERT models with all PEs generally attend more to preceding tokens than succeeding tokens, and this phenomenon appears to be stronger in learnable sinusoidal RPEs than others.

**The fully learnable APE and** [CLS] Fully learnable APE generally performs worse in translation invariance (see the 4-th column) as it has to deal with the unshiftable [CLS] which is always in the first position. Without considering [CLS] and [SEP] (see the 5-th column), the *fully learnable APE* satisfies translation invariance better than other APEs, showing that the *fully learnable APE* can flexibly deal with both special tokens and normal positions. The *fully learnable APE* also could handle the mismatch between special tokens and normal positions in the monotonicity property.

## 5 PEs in downstream tasks

We empirically compare the performance of PEs in classification and span prediction tasks.

**Fine-tuning** The fine-tuning on GLUE and SQuAD is the same as in the Huggingface website as per Wolf et al. (2019), see App. E for details. We report the average values of five runs per dataset. For classification, we use the GLUE (Wang et al., 2018) benchmark, which includes datasets for both single document classification and sentence pair classification. For span prediction, we use the SQuAD V1.1 and V2.0 datasets consisting of 100k crowdsourced question/answer pairs (Rajpurkar et al., 2016). Given a question and a passage from Wikipedia containing the answer, the task is to predict the answer text span in the passage. In V2.0, it is possible that no short answer exists in the passage since it additionally has 50,000 unanswerable questions written adversarially by crowdworkers (Rajpurkar et al., 2018).

### 5.1 Experimental results for downstream tasks

**GLUE** Tab. 3 shows that the fully-learnable APE (a.k.a, BERT-style APE) performs well in GLUE. No PE variants, especially BERT with solely APEs or RPEs, notably outperform the fully-learnable APE. BRRT models with a combination of an APE and an RPE do not always boost the performance of the model with solely the APE or RPE.

**SQuAD** Tab. 4 shows that nearly all BERT models with RPEs significantly outperform the *fully learnable APE*. The learnable sinusoidal APE is slightly better than the *fully learnable APE* in most cases. Both the best-performed models in SQuAD V1.1 and V2.0 adopt the fully-learnable RPE.

---

[10] 'monotonicity ' (second column) refers to monotonicity calculating in all relative distance, while 'monotonicity (first 20 offsets)' (third column) refers to monotonicity calculating within a relative distance of 20 (see App. C for monotonicity in other offsets.); the later matters since neighboring words in a small window are crucial in natural language. For translation invariance, we also adopt a new indicator without considering special tokens ([CLS] and [SEP]) to measure a purely position-aware translation invariance.

Table 3: Experiments on GLUE. The evaluation metrics are following the official GLUE benchmark (Wang et al., 2018). The best performance of each task is bold.

| PEs | single sentence | | sentence pair | | | | | | | |
|---|---|---|---|---|---|---|---|---|---|---|
| | CoLA acc | SST-2 acc | MNLI acc | MRPC F1 | QNLI acc | QQP F1 | RTE acc | STS-B spear. cor. | WNLI acc | mean ± std |
| BERT without PE | 39.0 | 86.5 | 80.1 | 86.2 | 83.7 | 86.5 | 63.0 | 87.4 | 33.8 | 76.6 ± 0.41 |
| fully learnable (BERT-style) APE | 60.2 | **93.0** | 84.8 | 89.4 | 88.7 | 87.8 | **65.1** | **88.6** | 37.5 | 82.2 ± 0.30 |
| fixed sin. APE | 57.1 | 92.6 | 84.3 | 89.0 | 88.1 | 87.5 | 58.4 | 86.9 | 45.1 | 80.5 ± 0.71 |
| learnable sin. APE | 56.0 | 92.8 | 84.8 | 88.7 | 88.5 | 87.7 | 59.1 | 87.0 | 40.8 | 80.6 ± 0.29 |
| fully-learnable RPE | 58.9 | 92.6 | 84.9 | **90.5** | 88.9 | **88.1** | 60.8 | 88.6 | 50.4 | 81.7 ± 0.31 |
| fixed sin. RPE | 60.4 | 92.2 | 84.8 | 89.8 | 88.8 | 88.0 | 62.9 | 88.1 | 45.1 | 81.8 ± 0.53 |
| learnable sin. RPE | 60.3 | 92.6 | **85.2** | 90.3 | **89.1** | **88.1** | 63.5 | 88.3 | 49.9 | 82.2 ± 0.40 |
| fully learnable APE + fully-learnable RPE | 59.8 | 92.8 | 85.1 | 89.6 | 88.6 | 87.8 | 62.5 | 88.3 | **51.5** | 81.8 ± 0.17 |
| fully learnable APE + fixed sin. RPE | 59.2 | 92.4 | 84.8 | 89.9 | 88.8 | 87.9 | 61.0 | 88.3 | 48.2 | 81.5 ± 0.20 |
| fully learnable APE+ learnable sin. RPE | **61.1** | 92.8 | **85.2** | 90.5 | 89.5 | 87.9 | **65.1** | 88.2 | 49.6 | **82.5** ± 0.44 |
| learnable sin. APE + fully-learnable RPE | 57.2 | 92.7 | 84.8 | 88.9 | 88.5 | 87.8 | 58.6 | 88.0 | 51.3 | 80.8 ± 0.44 |
| learnable sin. APE + fixed sin. RPE | 57.6 | 92.6 | 84.5 | 88.8 | 88.6 | 87.6 | 63.1 | 87.4 | 48.7 | 81.3 ± 0.43 |
| learnable sin. APE + learnable sin. RPE | 57.7 | 92.7 | 85.0 | 89.6 | 88.7 | 87.8 | 62.3 | 87.5 | 50.1 | 81.4 ± 0.33 |

Table 4: Performance (average and standard deviation in 5 runs) on *dev* of SQuAD V1.1 and V2.0. $\dagger$ indicates stat. significance over *fully learnable APEs* using a two-sided test with $p$-value 0.05.

| PEs | SQuAD V1.1 | | SQuAD V2.0 | |
|---|---|---|---|---|
| | F1 | EM | F1 | EM |
| BERT without PE | 36.47 ± 0.19 | 24.24 ± 0.33 | 50.48 ± 0.12 | 49.30 ± 0.14 |
| fully learnable (BERT-style) APE | 89.44 ± 0.08 | 81.92 ± 0.11 | 76.43 ± 0.63 | 73.07 ± 0.63 |
| fixed sin. APE | 89.45 ± 0.07 | 81.93 ± 0.11 | 76.12 ± 0.48 | 72.75 ± 0.55 |
| learnable sin. APE | 89.65$^\dagger$ ± 0.11 | 82.24$^\dagger$ ± 0.17 | 77.24 ± 0.43 | 73.93 ± 0.44 |
| fully-learnable RPE | 90.50$^\dagger$ ± 0.08 | 83.38$^\dagger$ ± 0.11 | 79.85$^\dagger$ ± 0.27 | 76.68$^\dagger$ ± 0.49 |
| fixed sin. RPE | 90.30$^\dagger$ ± 0.07 | 83.24$^\dagger$ ± 0.08 | 78.76$^\dagger$ ± 0.29 | 75.38$^\dagger$ ± 0.28 |
| learnable sin. RPE | 90.45$^\dagger$ ± 0.11 | 83.49$^\dagger$ ± 0.14 | 79.40$^\dagger$ ± 0.37 | 76.14$^\dagger$ ± 0.33 |
| fully learnable APE + fully-learnable RPE | 90.57$^\dagger$ ± 0.04 | 83.45$^\dagger$ ± 0.10 | **80.31**$^\dagger$ ± 0.10 | 76.94$^\dagger$ ± 0.20 |
| fully learnable APE + fixed sin. RPE | 90.24$^\dagger$ ± 0.17 | 83.06$^\dagger$ ± 0.21 | 78.74$^\dagger$ ± 0.50 | 75.40$^\dagger$ ± 0.52 |
| fully learnable APE+ learnable sin. RPE | 89.56 ± 0.28 | 82.26$^\dagger$ ± 0.30 | 77.82$^\dagger$ ± 0.42 | 74.51$^\dagger$ ± 0.39 |
| learnable sin. APE + fully-learnable RPE | **90.72**$^\dagger$ ± 0.13 | **83.68**$^\dagger$ ± 0.27 | 80.24$^\dagger$ ± 0.35 | **76.98**$^\dagger$ ± 0.34 |
| learnable sin. APE + fixed sin. RPE | 90.36$^\dagger$ ± 0.08 | 83.25$^\dagger$ ± 0.10 | 78.81$^\dagger$ ± 0.33 | 75.71$^\dagger$ ± 0.28 |
| learnable sin. APE + learnable sin. RPE | 90.49$^\dagger$ ± 0.14 | 83.59$^\dagger$ ± 0.14 | 79.93$^\dagger$ ± 0.34 | 76.69$^\dagger$ ± 0.39 |

As demonstrated in Tab. 2, the fully learnable APE can flexibly deal with `[CLS]` and translation invariance in normal positions, thus it performs well in classification tasks (GLUE) which heavily relies on the unshiftable `[CLS]` token for inference. Span prediction tasks which do not infer from `[CLS]` can benefit from strict translation invariance during parameterization (e.g., sinusoidal APEs and RPEs), see Tab. 5 in Sec. 6.1 for the correlations between performance of SQuAD and the translation invariance property. Removing PEs (*BERT without PE*) dramatically decreases performance in SQuAD V1.1 and V2.0, and slightly harms performance on GLUE, showing that PEs are more important in SQuAD than GLUE.

**Learnable sinusoidal PEs**    The sinusoidal APEs outperform fully-learnable APE in span prediction but underperform it in classification tasks. The learnable sinusoidal APE/RPE outperforms fixed sinusoidal APE/RPE in GLUE and SQuADs, showing the expressive power of flexible frequencies.

**Complementarity of APEs and RPEs**    In SQuAD, jointly adopting APEs and RPEs can slightly boost performance in some cases. For instance, BERT with *learnable sinusoidal APE + APE + fully RPE* achieves the best EM score in both SQuADs. However, this complementary effect is relatively weaker in GLUE, where the *fully-learnable APE* performs strongly.

# 6 DISCUSSIONS ON PEs

## 6.1 HOW DO THE PROPERTIES CORRELATE TO INDIVIDUAL TASKS?

We conduct a correlation analysis between the properties and the performance on individual tasks [11], as shown in Tab. 5. The results show that *violating monotonicity in relatively-small offsets (e.g., 20) and translation invariance is harmful* since it is negatively correlated to the performance on

---

[11]Pearson correlations are calculated between the property indicators and the performance of each individual task for 12 PEs. *BERT without PE* was not considered, since its property indicators are significantly different with other PEs and its performance is much worse; it therefore unexpectedly increases correlation values.

Table 5: Pearson correlations between the properties and evaluated tasks, evaluating on BERT models with 13 position embeddings. The positive (negative) numbers denote to which degree the performance of the task positively (negatively) correlate(s) to violating the property. This shows that *violating local monotonicity and translation invariance is harmful, while violating symmetry (and direction-balance) is beneficial.* Best correlation values are in bold for each row.

| Properties | | CoLA | SST-2 | MNLI | QQP | GLUE | SQuAD V1.1 | SQuAD V2.0 |
|---|---|---|---|---|---|---|---|---|
| monotonicity | all offsets | 0.44 | 0.43 | **0.56** | 0.32 | 0.48 | -0.31 | -0.27 |
| | first 20 offsets | -0.18 | 0.44 | -0.24 | -0.42 | -0.21 | **-0.91** | -0.86 |
| translation invariance | w/ [CLS]/[SEP] | 0.48 | 0.52 | 0.04 | -0.07 | 0.42 | **-0.63** | -0.57 |
| | w/o [CLS]/[SEP] | -0.47 | 0.01 | **-0.69** | -0.68 | -0.61 | -0.51 | -0.58 |
| symmetry | | 0.17 | 0.24 | **0.40** | 0.09 | 0.31 | 0.15 | 0.16 |
| direction balance | | 0.32 | 0.16 | **0.63** | 0.35 | 0.48 | 0.32 | 0.37 |

GLUE and SQuAD. However, *violating symmetry (and direction-balance) is slightly beneficial.* This shows that many tasks require BERT models to distinguish preceding and succeeding tokens, especially to attend more on preceding tokens. See Fig. 5b in App. C, the correlations between the direction balance indicators and the performance of downstream tasks will be much higher when only considering a few neighboring tokens for calculating the indicator.

## 6.2 MORE DISCUSSIONS ON THE PROPOSED PROPERTIES

**Monotonicity** Monotonicity holds locally in a small neighboring window (usually in 5-20 offsets) for all PE variants, see Fig.2. This shows that BERT models generally are not sensitive to longer-distance attendance patterns, also evidenced by the fact that performance in downstream tasks correlates more highly with monotonicity in middle-distance offsets (e.g., 20 in the second row of Tab. 5) than longer offsets (see App. C). To check monotonicity guided by learned frequencies of learnable sinusoidal APEs in individual tasks, see App. A.3

**Translation invariance** In BERT, we argue that absolute positions of words are uninformative since (1) absolute positions of the second segment depend on the length of the first sentence; (2) words are randomly truncated in the beginning or end if a sentence exceeds the expected maximum length, which may shift absolute positions of all tokens with an unexpected offset (Devlin et al., 2018). That is, absolute positions of words in pre-trained language models are arbitrarily replaceable, and thus adopting translation invariance is generally reasonable. Models with strict Translation invariance (all RPEs and sinusoidal APEs) naturally make PEs generalize to longer documents than the documents used in the pre-training phase, see App. F for some empirical evidence.

**Symmetry** APEs (especially sinusoidal APEs) express symmetry patterns without distinguishing the direction as shown in Fig 2. As seen from Eq. 7, it is nontrivial to model directions in two linearly-transformed query vectors and key vectors. This limits its performance in direction-sensitive downstream tasks. RPEs could behave better on direction perception, since forward and backward relative embeddings are separately embedded (see Tab. 1); Especially, learnable sinusoidal RPE or combination variants including it have more unbalanced attending patterns (see the last column in Tab. 2), as shown in Fig. 2 (f) and (h),

## 7 CONCLUSION

To theoretically and empirically understand position embeddings (PEs), we have defined three properties (translation invariance, monotonicity, and symmetry) inspired by distance mappings between the original domain of positions in $\mathbb{N}$ and their PEs in $\mathbb{R}^D$. A probing test has been proposed to quantitatively examine these properties using appropriate mathematical indicators. Our probing test has shown that these PEs nearly satisfy most properties even when they are fully-learnable without constraints. Experimental results have shown that violating local monotonicity and translation invariance decreases performance in downstream tasks (classification and span prediction tasks), and that violating symmetry benefits downstream tasks because of direction awareness. We also find that the fully-learnable absolute PE in general results in better performance for classification, and that relative PEs result in better performance for span prediction tasks, which can be explained by the connections between their properties and task characteristics.

ACKNOWLEDGMENTS

The work is supported by the Quantum Access and Retrieval Theory (QUARTZ) project, which has received funding from the European Union's Horizon 2020 research and innovation programme under the Marie Skłodowska-Curie grant agreement No. 721321.

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

## A    UNDERSTANDING FREQUENCIES

### A.1    UNDERSTAND INDIVIDUAL FREQUENCIES

We argue in this paper that a learning schema for such frequencies will be useful in a sense it could adaptively adjust frequencies to meet different functions, see Fig. 3.

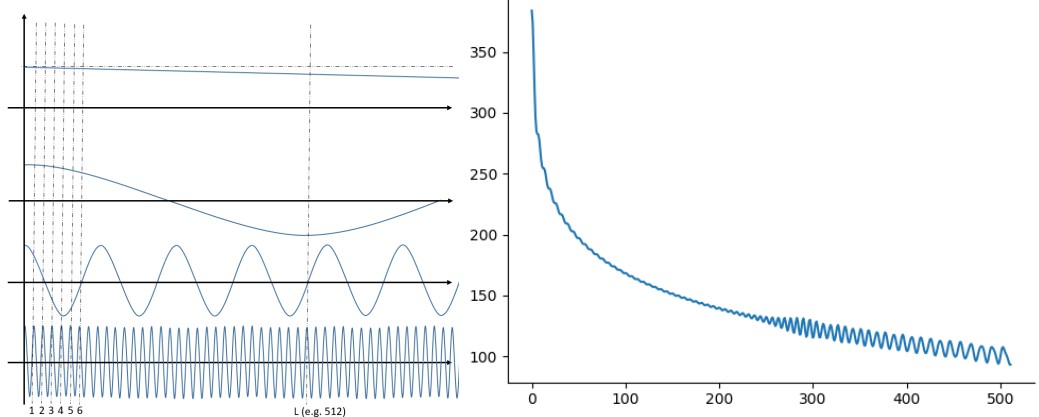

(a) Examples of some cosine functions    (b) $\phi(m)$, a sum of cosine functions with frequencies $\omega_i = (1/10000)^{2i/D}$.

Figure 3: $\phi(m)$ in (b) is a sum of many cosine functions of individual frequencies with increasing $m$, which determines the closeness between arbitrary two $m$-distance position vectors. As shown in (a), each frequency could play different roles: 1) the extremely small frequencies have few effects on the overall word representation ($\text{WE}_x + P_x$) in Eq. 5 since it makes such position embedding being almost identical with increasing positions; 2) some smaller frequencies can be beneficial to guarantee Property 1 if $\omega_i < \frac{\Pi}{L}$; 3) some bigger frequencies would promote the locally attending mechanism since such $\cos$ functions in Eq. 6 drop dramatically in the beginning if $\omega_i$ is great enough; 4) Some big frequencies which $\omega_i > \Pi$ would be smooth factors for the overall pattern since it would be randomly impose a bias to all positions.

### A.2    EXPRESSIVE POWER OF LEARNABLE SINUSOIDAL PES

In Transformers, linear transformation is commonly-used, for example *query*, *key*, and *value* transformations on word representations. Let $r_i$ be the word representation paramertezied by the sum of word embeddings and position embeddings (like the learnable sinusoidal APEs). Then, each element in $r_i$

$$r_{i,k}(t) = e_{i,k} + p_k(t) = \begin{cases} e_{i,k} + \sin(\omega_{\frac{k}{2}} t), & \text{if } k \text{ is even} \\ e_{i,k} + \cos(\omega_{\frac{k-1}{2}} t), & \text{if } k \text{ is odd} \end{cases} \tag{9}$$

After a linear transformation parameterized by $w$ (e.g., the key transformation $W^K$ in the first Transformer layer), $r_i$ is linearly transformed as $h_i(t) = w r_i$ ($h_i(t)$ can be one of query/key/value vectors $Q_x, K_x, V_x$ in $t$-th position) with each element

$$h_{i,k}(t) = \sum_{k=1}^{D} w_{j,k} e_{i,k} + \sum_{k=1}^{D/2} (w_{j,2k} \sin(\omega_{2k} t) + w_{j,2k+1} \cos(\omega_{2k+1} t)) \tag{10}$$

The RHS is a typical Fourier series with a base term $\sum w_{j,k} e_{i,k}$ and Fourier coefficients $\{w_{j,2k}, w_{j,2k+1}\}$. It is customarily assumed in physics and signal processing (Arfken & Weber, 1999) that the RHS in Eq. 10 with infinite $D$ and appropriate frequencies could approximate any continuous function on a given interval.

As using an infinite $D$ is not practical, dynamic allocation of a limited number of frequencies in a data-driven way could be beneficial for general approximation. The predefined frequencies $\omega_i =$

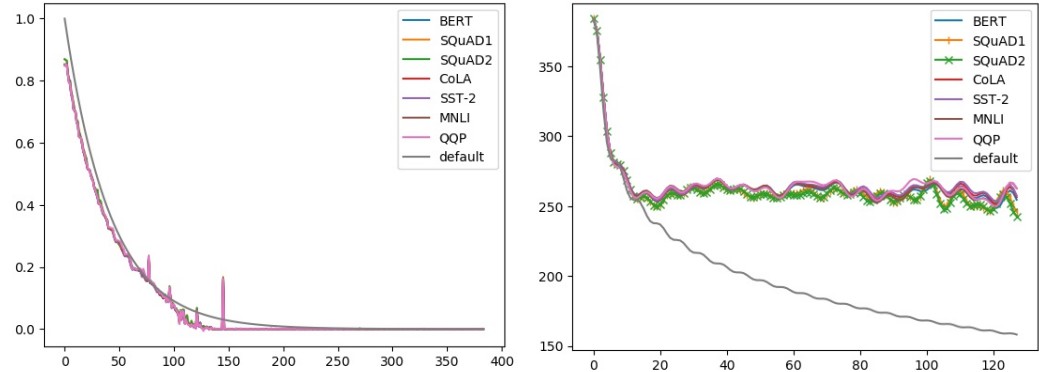

(a) The learned frequencies of pre-trained BERT and fine-tuned BERT in different downstream tasks. The predefined frequencies in (Vaswani et al., 2017) (i.e., $\omega_i = (1/10000)^{2i/D}$ ) is denoted as 'default'

(b) Dot products between two absolute positions with increasing offset, indicates neighboring APES are embedded together. $x$ axe refers to offset between positions.

Figure 4: The learned frequencies in *learnable sinusoidal APE* in the pre-trained language model and downstream tasks

$(1/10000)^{2i/D}$ in the Transformer (Vaswani et al., 2017) can be considered as a special case when it enumerates various frequencies ranging from $1/10000$ to $1$ under a specific distribution.

### A.3 LEARNED FREQUENCIES OF LEARNABLE SINUSOIDAL APE

The learned frequencies are shown in Fig. 4a. Observe that the learned frequencies are generally smaller than the pre-defined ones from (Vaswani et al., 2017) (i.e., $\omega_i = (1/10000)^{2i/D}$). The learned frequencies are close to the learned one since we use $\omega_i = (1/10000)^{2i/D}$ as initialization.

As shown in Fig. 4b, the patterns of dot products between positions for learnable frequencies are quite different to the predefined ones (denoted as 'default' in the figure); indeed, the former appears more predisposed to deeming remote positions similar. Moreover, fine-tuned models for span prediction tasks (including SQuAD and SQuAD2) satisfy strict monotonicity in larger windows than for classification tasks. Observe also that the patterns in pre-training language models seem more similar to those in classification tasks than span prediction tasks.

## B QUANTITATIVELY MEASURING THE PROPERTIES.

To quantitatively measure the primary properties we treat in this paper, we propose multiple criteria, described below.

Assume a position-wise attention matrix $\bar{A}^{(1)}$ (denoted as $A$ since there is no risk for confusion), in which each element is the (softmax) activated attention value from the $i$-th query token to $j$-th key token (all elements are positive).

**Average In-group Variance (AIV) for translation invariance**  Let $l$ be an offset between two positions; we denote by $\tau(l)$ the set of $l$-offset attention values $\{A_{i,j}, j - i = l\}$; for example, $\tau(1) = \{A_{1,2}, A_{2,3}, \cdots, A_{L-1,L}\}$. Translation invariance requires that all elements in each group $\tau(l)$ should be identical, the smaller variance each $\tau(l)$ has, it is closer to translation invariance. The *Average In-group Variance* (AIV) is defined as a weighted average over in-group variances of all $\{\tau(l)\}_{-L+1}^{L-1}$, namely:

$$\text{AIV}(A) = \frac{\sum_{l=-L+1}^{L-1} \text{var}\left(\tau(l)\right) \cdot |\tau(l)|}{\sum_{l=-L+1}^{L-1} |\tau(l)|}$$

where $|\cdot|$ is the number of elements in the set. For normalization, this metric is further divided by the overall variance (i.e., $\text{var}(A)$).

**Ordered Pair Ratio (OPR) for monotonicity**    For a word in $i$-th position, based on the increasing distance to the $i$-th position, there a forward attention sequence $S_{i,+} = \{A_{i,i}, A_{i,i+1}, \cdots, A_{i,L}\}$ and a backward attention sequence $S_{i,-} = \{A_{i,i}, A_{i,i-1}, \cdots, A_{i,1}\}$. This results in $2L$ sequences denoted as $\mathbb{S} = \{S_{1,+}, S_{1,-}, S_{2,+}, S_{2,-}, \ldots, S_{L,+}, S_{L,-}\}$. The ideal (decreasing) monotonicity requires that each $S$ (an element in $\mathbb{S}$) is totally ordered as $s_0 > s_1 > \cdots > s_{L-1}$. We define the *Ordered Ratio* of $S$ by:

$$\text{OPR}(S) = \frac{\sum_{s_j, s_i \in S, i \neq j} \text{sign}\left((s_i - s_j)(i - j)\right)}{|S|^2 - |S|}$$

We define $\text{sign}(x) = 1$ if $x > 0$, and $\text{sign}(x) = 0$ otherwise. Ideally, the OPR of a totally ordered decreasing (increasing) sequence $S$ should be zero (one). The expected OPR of a randomly-ordered sequence (average OPR of the set of all such sequences) should be 0.5.

Finally, we get a weighted sum of OPRs of all sequences in $\mathbb{S}$.

$$\text{OPR}(A) = \frac{\sum_{S \in \mathbb{S}} \text{OPR}(S) \cdot |S|}{\sum_{S \in \mathbb{S}} |S|}$$

In the paper, we also consider a version of monotonicity within a offset of $k$ (e.g., 'monotonicity (first 20 offsets)' in Tab. 2), which OPR is calculated in first $k$ elements of each $S \in \mathbb{S}$.

**Symmetrical Discrepancy for symmetry**    We define the *Symmetrical Discrepancy* (SD) by:

$$SD(A) = \frac{\sum_{i,j,i<j} |A_{i,j} - A_{j,i}|}{L \times (L-1)/2}$$

**Direction Balance**    We define the *Direction Balance* (DB) as the ratio between the sum of the lower (left) triangle and the upper (right) triangle of $A$. Note that all elements are positive in $A$, DB(A) in $l$-offset range is always positive.

$$DB_l(A) = \frac{\sum_{i,j;i<j,|i-j|<=l} A_{i,j}}{\sum_{i,j;i>j,|i-j|<=l} A_{i,j}}$$

In Tab. 5 and 2, we report $DB$ for a offset range of 20, see Fig. 5b for the performance correlations with other offset ranges.

## C    MEASURING CORRELATIONS BETWEEN PROPERTIES AND DOWNSTREAM TASKS.

In Tab. 5, monotonicity in 20 offsets and the correlations with performance in downstream tasks was reported; here we show how different ranges of the monotonicity correlate to performance in downstream tasks. Among all tasks, we choose all single sentence classification tasks (CoLA and SST-2), two biggest sentence pair classification tasks (MNLI and QQP tasks have more training samples than others), average performance in GLUE, and in SQuAd (F1 metrics nearly have identical trends with EM metrics).

As shown in Fig. 5a, the monotonicity indicators in nearly 20-55 offsets are highest correlated to the performance of span prediction tasks (with Pearson correlation larger than $90\%$). Note that some classification tasks (especially SST-2) also show opposite correlations comparing to span prediction tasks, probably due to the unshiftable `[CLS]` on which classification tasks rely for interference does not need monotonicity.

In Fig. 5b, the performance correlates more to the direction balance indicators when considering neighboring tokens. For instance, the direction balance indicators within a small offset has correlation bigger than 0.5, this tends to be smaller with increasing offsets.

## D    RELATIVE POSITION EMBEDDING WITH LONG OFFSETS

The dot product between two position embeddings are shown in Fig. 1(e) and (f). To analyze the behaviour, we replace the raw dot product with the cosine similarity (as the latter is normalized and

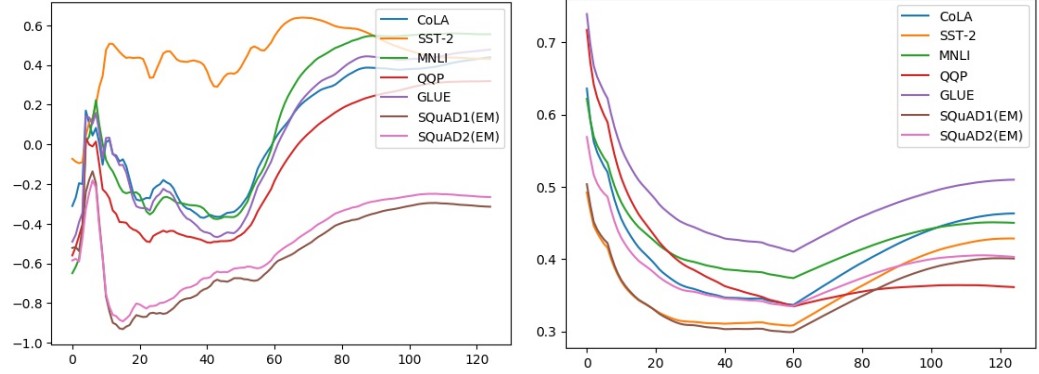

(a) Pearson correlations between the performance of downstream tasks (shown in Tables 3 and 4) and **monotonicity** indicators in different offset ranges. This shows *violating monotonicity (especially in a 15-60 offset range) is harmful for most tasks.*

(b) Pearson correlations between the performance of downstream tasks (shown in Tables 3 and 4) and **direction balance** indicators in different ranges. This shows the *more attending to preceding tokens than succeeding tokens (especially for neighboring tokens) usually leads to better performance for GLUE in SQuAD.*

Figure 5: In which offset ranges the properties correlate with the performance in downstream tasks.

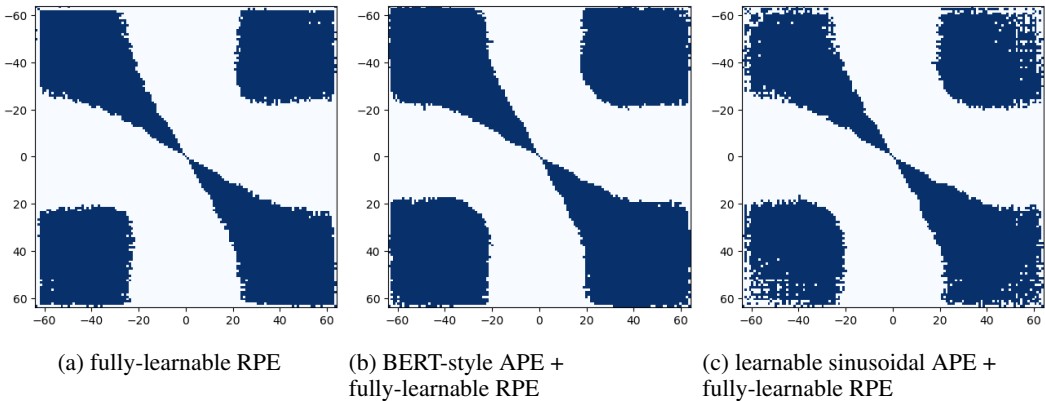

(a) fully-learnable RPE

(b) BERT-style APE + fully-learnable RPE

(c) learnable sinusoidal APE + fully-learnable RPE

Figure 6: Cosine similarities between any two relative position vectors. Cosine similarities bigger than 95% are in blue.

thus easier to interpret). When the cosine similarity is one, the two vectors are perfectly colinear and share the same direction. For the purposes of this investigation, we arbitrarily pick 0.95 as a threshold for the cosine similarity, denote that two vectors are not significantly different.

From Fig. 6 we observe the following for all PE variants with fully-learnable RPE: (1) There is no significant difference between relative position vectors with longer than 20-25 offsets; (2) forward relative position vectors are slightly more similar to forward relative position vectors instead of backward relative position vectors, and vice versa (see the central left-lower/right-upper white parts).

## E    DETAILED EXPERIMENTAL SETTING

We train BERT base and BERT medium with both masked language prediction and next sentence prediction tasks; most parameters are listed in Tab. 6, with the remaining parameters set as in the original paper. Note that we share RPE in different heads and layers. Like (Shaw et al., 2018) RPE are truncated from $-64$ to $64$.

Table 6: Detailed Experimental Settings

| Training | pre-training from scratch | max Length | epoch | learning rate | batch size |
|---|---|---|---|---|---|
| BERT-base on 128 length | ✗ | 128 | 5 | 5e-5 | 64 |
| BERT-base on 512 length | ✗ | 512 | 2 | 5e-5 | 512 |
| BERT-medium on 128 length | ✓ | 128 | 10 | 5e-5 | 128 |
| BERT-medium on 512 length | ✓ | 512 | 2 | 5e-5 | 512 |
| GLUE | - | 128 | 3 | 2e-5 | 32 |
| SQuAD | - | 384 | 3 | 3e-5 | 32 |

We perform five runs for SQuAD and GLUE benchmark. The results in GLUE are for the last checkpoint during fine-tuning while SQuAD takes the best one for every 1000 steps. Finally, we calculate the average over 5 runs. All these settings are the same for all PEs. We use Mismatched MNLI. In GLUE (Wang et al., 2018), the train and dev are somewhat adversarial: training samples (in train and dev) containing the same sentence usually have opposite labels. Models may get worse when it overfits in the train set, resulting in unexpected results. Therefore, we exclude WNLI to calculate average in the last column in Tab. 3. The fine-tuning parameters are using default values in Huggingface project Wolf et al. (2019).

## F  GENERALIZATION TO LONGER SENTENCES IN DOWNSTREAM TASKS

To fairly compare all models, we train a *medium* setting (8-layer transformer) on 128-length input in the first 10 epochs and 512-length input in the last 2 epochs from scratch. Fig. 7 shows that before 512-length pre-trained (like the 10-th epoch 128-length pre-trained) *learnable sinusoidal APEs* and RPEs perform better than BERT-style (without sinusoidal parameterization) in both SQuADs. This happens because PEs with translation invariance (*learnable sinusoidal APEs* and RPEs) generalize into longer positions [12], while position vectors between 128-512 positions are not trained in fully-learnable PEs and they are randomly initialized and finetuned in the downstream.

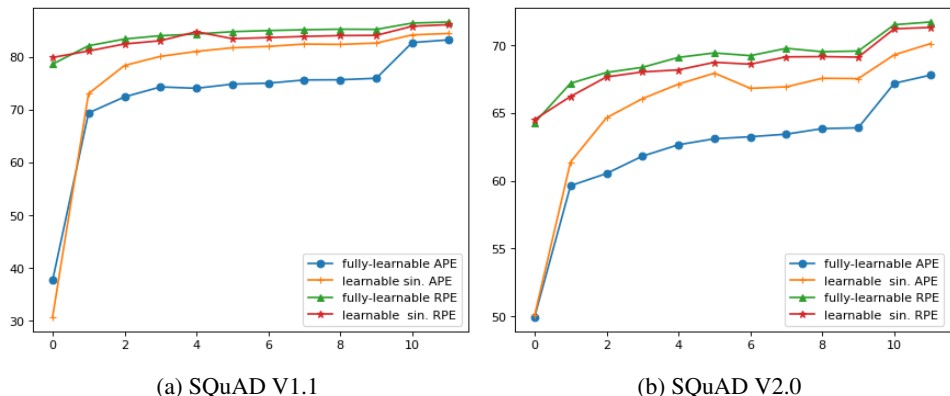

(a) SQuAD V1.1                    (b) SQuAD V2.0

Figure 7: Experimental results on SQuADs with *BERT-medium*. X-axis: epoch number (first trained on 128-length seq. with 10 epochs and then 512-length with 2 epochs). Y-axis: F1 score.

## G  THE EVOLUTION OF DOT PRODUCTS BETWEEN POSITION VECTORS

We exhibit dot products between position vectors during training a BERT-medium, as shown in Fig. 8. There is seemingly no pattern in the beginning, but as the number of training steps increase, a regular pattern with translation invariance and local monotonicity emerges.

---

[12]In practice, the document length of some tasks, like summarization, document-level translation, etc. may be much longer than the maximum length typical BERT models can deal with, i.e., 512. Then, learnable sinusoidal PEs or RPEs would be beneficial. Note that they also save parameters compared to typical BERT models, especially when document length is very long like (Beltagy et al., 2020).

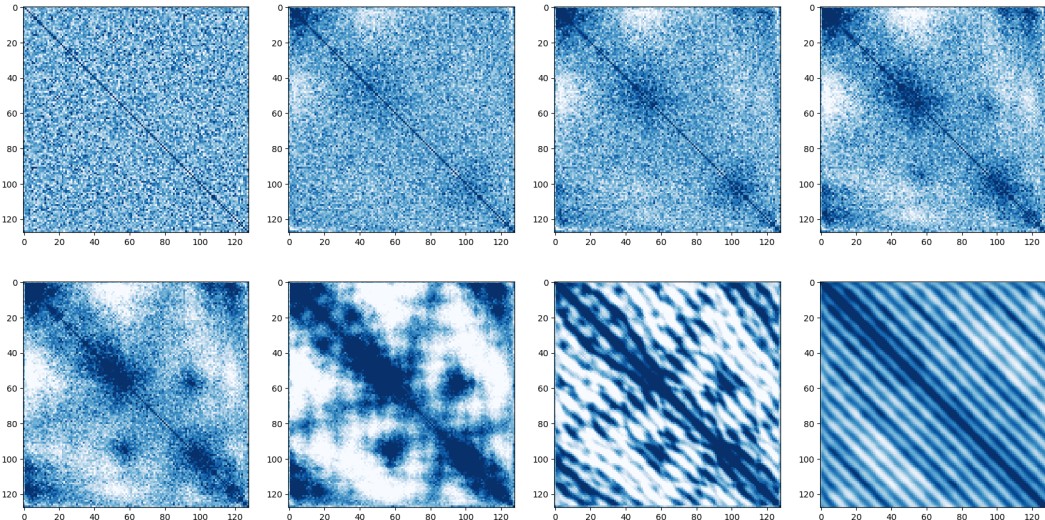

Figure 8: Dot products between absolute position vectors evolving with training steps.

## H    DISCUSSIONS ON RELATED WORKS

**Complementary effect between APE and RPE**    The complementary effect between APE and RPE was demonstrated to be effective in (Wang et al., 2019) for machine translation. In the pre-trained language model, Ke et al. (2020) propose that combining APE and RPE could be beneficial for classification tasks (GLUE), which in this paper, this complementary effect is not significant since most PE combinations (APE and RPE) do not outperform the BERT-style fully-learnable APE on classification. Instead, we empirically conclude that most PE combinations boost the performance in span prediction tasks. The benefit in classification tasks in (Ke et al., 2020) may come from other modifications, for example, it unties the [CLS] symbol from other positions. Moreover, in the paper, it adopts a special relative position embedding like (Raffel et al., 2019) (as this paper also suggests to do so): a simplified form of PE that each "embedding" is simply a scalar bias added to the corresponding logits when computing the attention weights. The fundamental difference between the 'position bias' and position embedding is unknown from now.

**Study on attention visualization.**    Many works are focusing on understanding attention patterns in individual heads. For example Vig (2019) introduced a tool for visualizing attention in the Transformer at multiple scales; Rogers et al. (2020) suggest attention mechanisms like Vertical, Diagonal, Vertical + diagonal, Block, and Heterogeneous. Clark et al. (2019) found some attention mechanisms like attending broadly, to next, to [CLS] or [SEP], attend to punctuation. While our paper focuses on the general attention introduced by PEs from an average point of view, without considering any specific attention head.

**Asymmetry in sequential labeling**    Yan et al. (2019) suggested asymmetry of position embedding in named-entity recognition task (without involving pre-trained language models) which is a kind of sequential labeling tasks like span prediction (SQuAD) in this paper. Their conclusion is generally compatible with ours, but we question its assumption that 'the property of distance-awareness disappears when query and key projection are conducted'. As shown in Fig. 9, we could slightly see some distance-awareness by directly taking the average position-position correspondence in the first layer among many heads (i.e., $PW^{Q,1}(W^{K,1})^T P^T$).

**Functional parameterization of PEs**    Xu et al. (2019) proposes various variants of sinusoidal positional encodings inspired by functional analysis. Wang et al. (2020) proposed a sinusoid-like complex word embedding to encode word order. Both (Xu et al., 2019) and (Wang et al., 2020) assume that PEs should satisfy the translation invariance property, but they induce different types of sinusoidal PE parameterization either in real or complex vector space. Moreover, Liu et al. (2020) use a neural ODE component to parameterize position encoding as a continuous dynamical model, which could learn suitable PEs in neural networks. All of these PEs are inspiring. Since selecting

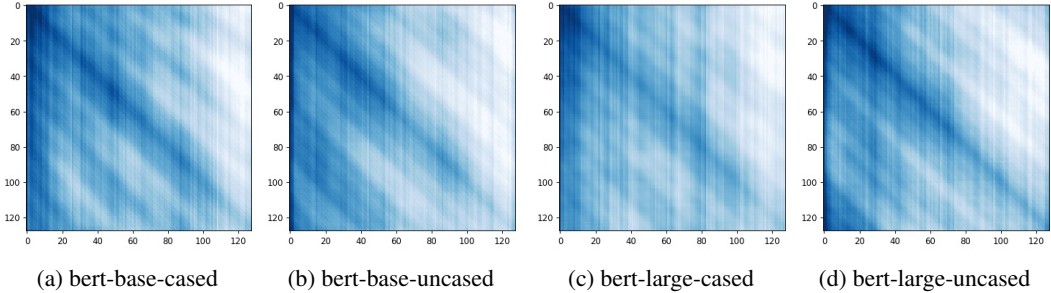

| (a) bert-base-cased | (b) bert-base-uncased | (c) bert-large-cased | (d) bert-large-uncased |

Figure 9: Position-wise correlation matrix $(PW^{Q,1}(W^{K,1})^T P^T)$ for first 128 positions in BERT pre-trained models

the suitable parameterization type is not the main concern in this paper, we adopted the typical ones, namely, the fully-learnable, (learnable or fixed) sinusoidal APEs/RPEs. The fundamental difference between these PE parameterizations needs further investigation. More recently, (Wang & Chen, 2020) empirically study the behaviour of many position embeddings and their performance in Transformers for various NLP tasks.

## I  THE THREE PROPERTIES IN OTHER MODELS

By using the proposed identical probing test, we also check the properties of other trained Transformer models with decoder components in Tab. 7 and Fig. 10. The machine translation model [13] is a typical encoder-decoder architecture using multiple-layer Transformers. GPT2 (Radford et al., 2019) adopts a purely decoder architecture; 12-layer *base* setting is used in this work.

**Monotonicity**   Compared to BERT and the machine translation, GPT2 satisfies monotonicity (especially in the first 20 offsets) better than other models, showing capturing distance between neighboring tokens matters in the language model.

**Translation invariance**   As seen from the translation invariance indicators in Tab. 7, GPT2 satisfies translation invariance poorer than other models, since tokens in it also additionally attend to a few beginning tokens no matter how far the attended tokens are.

**Symmetry**   GPT2 shows the biggest symmetrical discrepancy, since GPT2, which aims to predict the next word, adopts an attention mask of succeeding tokens to avoid information leakage. Plus, the machine translation encoder slightly attends more to the succeeding tokens while BERT attends more on the preceding tokens than succeeding tokens.

Table 7: Quantitative measurement of the properties for models of machine translation, language models.

| PEs | PE type | model type | monotonicity | | translation invariance w/o special tokens | symmetry | direction balance |
| --- | --- | --- | --- | --- | --- | --- | --- |
| | | | all offsets | first 20 offsets | | | |
| BERT | fully-learnable APE | encoder only | 0.2461 | 0.0208 | **0.0143** | 0.0012 | 1.1940 |
| GPT | fully-learnable APE | decoder only | **0.1019** | **0.0044** | 0.1114 | 0.0070 | inf |
| Machine Translation | fixed sin. APE | encoder & decoder | 0.3540 | 0.0841 | 0.0214 | **0.0002** | 0.8074 |

## J  WHITE BAND EFFECTS ALONG THE DIAGONAL

In order to analyze the *white band effects* along the diagonal, we show all results of identical word probing (average attention values in the first layer of identical word probing with respect to 100

---

[13]An English-to-French machine translation model downloaded from `https://github.com/pytorch/fairseq/tree/master/examples/translation` (Vaswani et al., 2017; Ott et al., 2018; 2019)

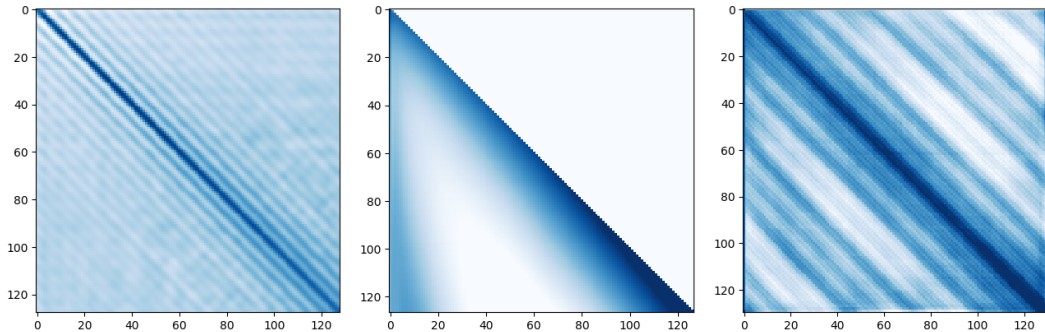

(a) Encoder (w/ Decoder): Machine (b) Decoder : Generative language (c) Encoder : Masked language translation (Vaswani et al., 2017) model (GPT2) (Radford et al., 2019) model (BERT) (Devlin et al., 2018)

Figure 10: Identical word probing with different types of trained models.

randomly-selected words). This effect is more clear for fully-learnable RPE, learnable sinusoidal RPE and any combination variants including them (see. Fig. 11 (d,f,g,i,j,l)). To show the obvious differences between these PEs, in this paper, we use average unnormalized attention weights matrix for probing, but all indicators are calculated using normalized attention values for better quantitative comparison.

## K  THE REPLACEABLE PROPERTY ABOUT ABSOLUTE POSITIONS OF WORDS

For example (we do not consider subword tokenization for simplicity), we have two sentences for next sentence predictions (As BERT did)

sentence1 : `Deadlines are the No.1 productive forces .`

sentence2 : `I think , therefore I am .`

By adding three special tokens, we will have a example with 17 tokens as

```
[CLS] Deadlines are the No.1 productive forces .  [SEP] I think ,
therefore I am .[SEP]
```

with absolute positions in the bracket as

```
[CLS](1) Deadlines(2) are(3) the(4) No.1(5) productive(6)
forces(7) .(8) [SEP](9) I(10) think(11) ,(12) therefore(13) I(14)
am(15) .(16) [SEP](17)
```

Assume that the expected maximum sequence length is 16 (actually 128 or 512 in BERT), we need to randomly remove the first token of the first sentence (i.e., `Deadlines` ) as

**valid sample: I:** `[CLS](1) are(2) the(3) No.1(4) productive(5) forces(6)`
`.(7) [SEP](8) I(9) think(10) ,(11) therefore(12) I(13) am(14)`
`.(15) [SEP](16)`

or last token of the second sentence (i.e., `.`  )

**valid sample:    II:**    `[CLS](1) Deadlines(2) are(3) the(4) No.1(5)`
`productive(6) forces(7) .(8) [SEP](9) I(10) think(11) ,(12)`
`therefore(13) I(14) am(15) [SEP](16)`

Both the above two sentences are valid for training. If we replaced the first sentence with another shorter sentence (i.e., Publish/Launch or Perish ?), the sample would be

**valid sample:    III:**    `[CLS](1) Publish(2) or(3) Perish(4) ?(5) [SEP](6)`
`I(7) think(8) ,(9) therefore(10) I(11) am(12) [SEP](13) [PAD](14)`
`[PAD](15) [PAD](16)`

The three samples I,II,III are valid, but its absolute position indexes are not shiftable. Especially, the first sentence of the second sentence could be 9, 10, and 7, respectively, depending on the random seed for dropping and the length of the first sentence.

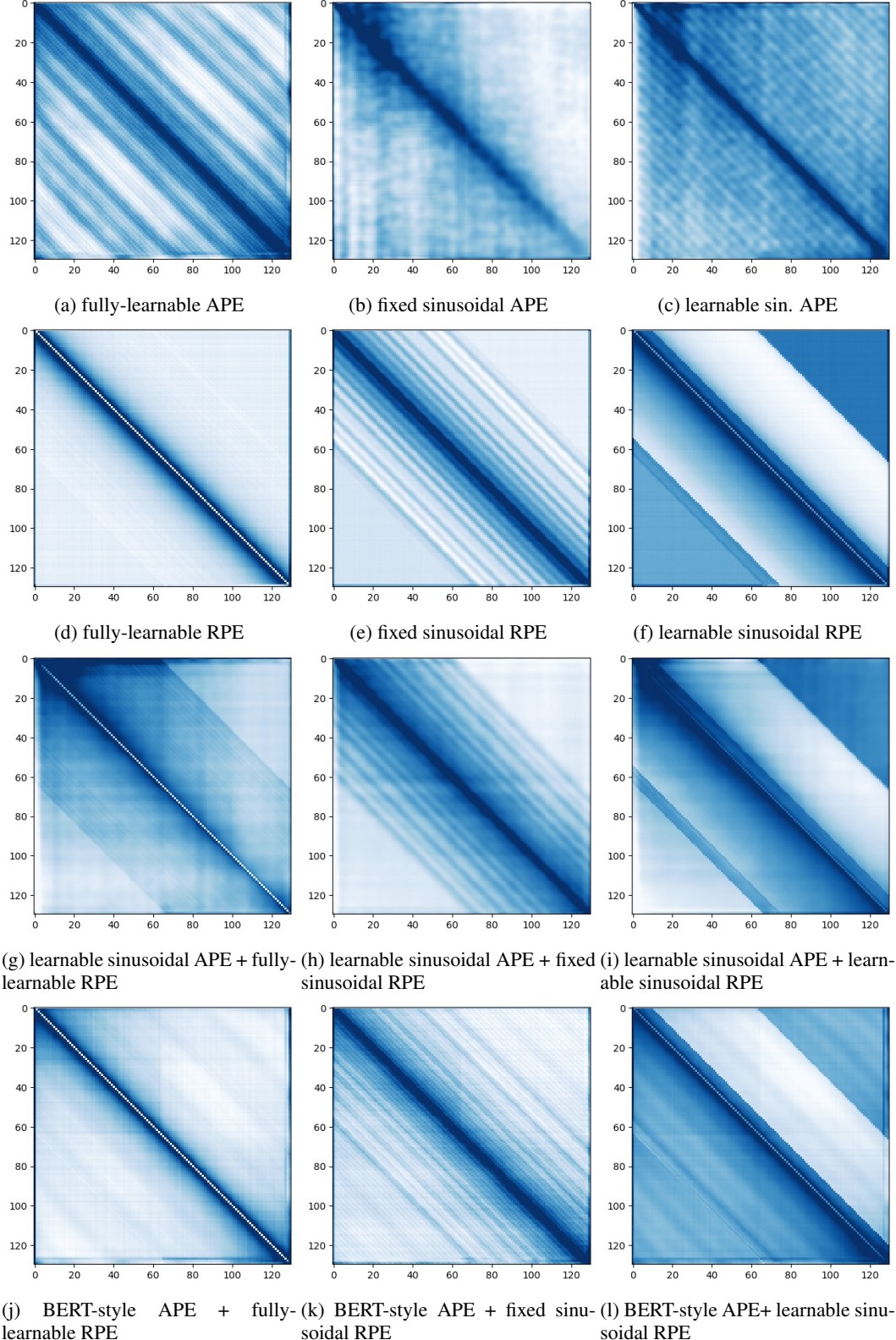

(a) fully-learnable APE

(b) fixed sinusoidal APE

(c) learnable sin. APE

(d) fully-learnable RPE

(e) fixed sinusoidal RPE

(f) learnable sinusoidal RPE

(g) learnable sinusoidal APE + fully-learnable RPE

(h) learnable sinusoidal APE + fixed sinusoidal RPE

(i) learnable sinusoidal APE + learnable sinusoidal RPE

(j) BERT-style APE + fully-learnable RPE

(k) BERT-style APE + fixed sinusoidal RPE

(l) BERT-style APE+ learnable sinusoidal RPE

Figure 11: Identical word probing (models with more PEs are shown here comparing to Fig. 2). Darker in the $i$-th row and $j$-th column means that the $i$-th words generally attend more on the $j$-th words.

