# OpenReview forum: "On Position Embeddings in BERT"
_ICLR.cc/2021/Conference — ICLR 2021 Poster_

### Official Review · AnonReviewer3 · 2020-10-26
**An empirical study work that lacks convincing quantitative analysis**

**Rating:** 6
**Confidence:** 5

**Review:**

This paper studies the position embeddings of transform-based models, and proposed a unified position embedding evaluation method in three aspects, i.e., Monotonicity, Translation invariance and Symmetry, which can well summarise the properties of the existing position embedding methods.

### Strengths of the paper

1. It is good that the authors summarise three property for position embedding models, and discuss four related position embedding models under the three properties.
2. The three properties proposed by this paper are suitable for most position embedding models.
3. The experimental details are complete and easy to reproduce. Extensive experiment details are provided in the appendix.

### Weaknesses of the paper

1. The presentation and organization of this paper should be improved. Typos and language issues can be easily found, see the minor comments below. The contributions are not well highlighted in both the abstract and introduction section.

2. The authors take many efforts to conduct experiments for the position embedding of BERTs, and provide an empirical study of the four existing position embedding models (fully learnable APEs (Gehring et al., 2017), (2) fixed sinusoidal APEs (Vaswani et al., 2017), (3) fully learnable RPEs (Shaw et al., 2018), and (4) fixed sinusoidal RPEs (Wei et al., 2019).). The author tested these models and their combinations over three benchmarking datasets, however, as an empirical study paper, the analysis is too weak.

(1) For the qualitative result shown in Table 2 and Table 3, neither insight nor connection to three properties is provided for the result. In fact, the result shown in Section 4 is intuitive and not something new, since these are the basic motivations of the RPEs and learnable PEs. Moreover, the effectiveness of using both RPE and APE has already been validated in "Self-Attention with Structural Position Representations, arXiv:1909.00383. 2019".

(2)  I would like to see, in the experiment part, some new experimental results and conclusions in terms of the four summarised properties, which are key contributions the authors claimed. However, what I see is just some position vector embedding similarities (i.e. Fig. 2 and Fig.3) in terms of the four position embedding models, where the results are also somehow expected and intuitive. The conclusions of the experimental results are too subjective. For example, from the analysis of Fig (2), the conclusion that : “Lastly, note that fully-learnable RPEs also do not significantly distinguish far-distant RPEs (from -64 to -20 and from 20 to 64), suggesting that truncating RPEs into a distance of 64, like (Shaw et al., 2018), is reasonable.”  is a bit farfetched. In this figure the white part is as narrow as (-5,+5),  a further quantitative evidence in other forms for this conclusion will be more preferable. The same issue also exists in the conclusion of Figure (3): “This may be an advantage of RPEs over APEs to perceive forward and backward words, especially in span prediction tasks where capturing this matters.”

3. This paper lacks the discussion with the latest position embedding models such as,

   [1] "Learning to Encode Position for Transformer with Continuous Dynamical Model". ICML. 2020

   [2] "Encoding Word Order in Complex Embeddings",  ICLR. 2019, where the 'Translation' property is also discussed.

Other minor comments:

1. In the introduction section, "distances in $\mathbb{N}$ and $\mathbb{R}^{D}$ ", $\mathbb{N}$ and $\mathbb{R}^{D}$ should be explained when they appear in the first place.

2. The references should be formatted in a unified manner.

3. Table 2, the bold indicators for the best performances are put on the wrong numbers, e.g. in QNLI, the bold should be 89.5,  in WNLI task it should be 51.3, and in STS-B it should be 87.5.


Overall, I think this paper indeed shows some interesting empirical results of position embedding models for BERT. But, the analysis is monotonous and too subjective, lacking the necessary mathematical quantitative indicators, which prevents it from being a general way to verify the conclusions in this paper.


------
### Comments after the discussion

Thank you for your detailed response. I think most of my concerns were addressed so I updated the score to 6.

---

> ### Author Response · Authors · 2020-11-23
> **In revision, we added quantitative  measurement for the three properties and  conduct statistical correlations between properties and performance in individual tasks**
>
>
> We thank you for the helpful comments and suggestions. We provide our responses as follows.
>
> **Weaknesses1:  The presentation and organization of this paper should be improved. Typos and language issues can be easily found, see the minor comments below. The contributions are not well highlighted in both the abstract and introduction section**
>
> We apologize for the typos and language issues; we have now fixed them.
> In the latest version, we have rewritten the introduction and abstract; all contributions are now listed in the introduction. In addition, the paper has been scoured for typos and language issues.
>
> **Weaknesses2: The authors take many efforts to conduct experiments for the position embedding of BERTs, and provide an empirical study of the four existing position embedding models (fully learnable APEs (Gehring et al., 2017), (2) fixed sinusoidal APEs (Vaswani et al., 2017), (3) fully learnable RPEs (Shaw et al., 2018), and (4) fixed sinusoidal RPEs (Wei et al., 2019)). The author tested these models and their combinations over three benchmarking datasets, however, as an empirical study paper, the analysis is too weak.**
>
> In the latest version, we have rewritten most of the paper devoted to the analysis of the experimental results, and we have added the following analyses:
>
> (1)  We have added quantitative  measurement for the three properties in Sec. 4.1.2 (Table 2) and Appendix B;
>
> (2) We have conducted an analysis of the correlations between properties and performance in individual tasks, in Sec. 4.3 (Table 4).
>
> (3)  We have investigated the difference between learned frequencies and predefined ones (Vaswani et al. 2017) in  Appendix A.3.
>
> (4) We have shown how models are insensitive to long-distance offsets in Appendix C.
>
> (5) We have investigated how PEs evolve with an increasing number of training steps in Appendix F, in particular, PEs gradually increasingly satisfy some of the three main properties we examine in the paper.
>
> (6)The white band effect indicates words do not attend themselves in learnable RPEs, as previously stated by Clark et al. 2019.
>
> (7)Comapring the position related attention patterns with GPT and a machine translation model in App. I

---

> ### Author Response · Authors · 2020-11-23
> **Reply to Reviewer3's comment**
>
> **Comment1:  For the qualitative result shown in Table 2 and Table 3, neither insight nor connection to three properties is provided for the result. In fact, the result shown in Section 4 is intuitive and not something new, since these are the basic motivations of the RPEs and learnable PEs. Moreover, the effectiveness of using both RPE and APE has already been validated in "Self-Attention with Structural Position Representations, arXiv:1909.00383. 2019".**
>
> We have added quantitative indicators for the three properties in Sec. 4.2 (Table 2), and have added a correlation study to quantitatively measure how these properties correlate with the performance of downstream tasks, in Sec. 6 (Table 5):  both classification and span prediction benefit from the translation invariance and local monotonicity properties of PEs, while symmetry slightly decreases their performance because of inability to perceive directions. See. Tab. 5 for more details.
>
> Also, we have added the suggested reference in App. H.
>
> **Comment2:  I would like to see, in the experiment part, some new experimental results and conclusions in terms of the four summarised properties, which are key contributions the authors claimed. However, what I see is just some position vector embedding similarities (i.e. Fig. 2 and Fig.3) in terms of the four position embedding models, where the results are also somehow expected and intuitive. The conclusions of the experimental results are too subjective. For example, from the analysis of Fig (2), the conclusion that : “Lastly, note that fully-learnable RPEs also do not significantly distinguish far-distant RPEs (from -64 to -20 and from 20 to 64), suggesting that truncating RPEs into a distance of 64, like (Shaw et al., 2018), is reasonable.” is a bit farfetched. In this figure the white part is as narrow as (-5,+5), a further quantitative evidence in other forms for this conclusion will be more preferable. The same issue also exists in the conclusion of Figure (3): “This may be an advantage of RPEs over APEs to perceive forward and backward words, especially in span prediction tasks where capturing this matters.”**
>
>
> 1. Other than position vector embedding similarities, we have added some results concerning a new probing test,  have added quantitative indicators to measure the properties, and have added quantitative correlations between the properties and downstream tasks.
>
> 2. We apologize that there were some subjective conclusions in the initial versions.  As suggested by you, we have (1) designed some quantitative indicators to measuring these properties; and (2) we have computed quantitative correlations between the properties and downstream tasks. All subjective arguments have been carefully checked and removed in the latest version.
>
>
>
> 3. We have revised the description of distinguishable far-distant RPEs by carefully stating the white parts and dark parts. The white parts (-5,+5) indicates the relative position vectors for small offsets (e.g., $\{p_{-5},\cdots, P_0, \cdots, P_{-5} \}$ are notably different to other relative position vectors.
> In Tab 6 of App. D, we checked the dark parts by adopting a normalized metric, i.e., cosine similarity between any two relative position vectors, the results show that RPE with big offsets (e.g., $\{P_{-64},\cdots,P_{-20}\}$ and $\{ P_{20},\cdots, P_{60}\}$) are very similar (with cosine similarities approaching or exceeding 0.95).
>
> 4. We have added two quantitative indicators regarding directions in the new version, and carefully revised the augment about `an advantage of RPEs over APE'.
>
>
>
> **Comment3: This paper lacks the discussion with the latest position embedding models such as Liu et al. ICML 2020 and Wang et al. ICLR 2020,**
>
>
> We have now added more discussion of the related work (Liu et al. ICML 2020 and Wang et al. ICLR 2020) in the App. H due to page limitations. We also discuss the recently-released paper by Wang and Yun-Nung  below.
>
> Wang, Yu-An, and Yun-Nung Chen. "What Do Position Embeddings Learn? An Empirical Study of Pre-Trained Language Model Positional Encoding." arXiv preprint arXiv:2010.04903 (2020). Accepted by EMNLP, publicly available after the ICLR deadline.
>
> **Comment4:  Overall, I think this paper indeed shows some interesting empirical results of position embedding models for BERT. But, the analysis is monotonous and too subjective, lacking the necessary mathematical quantitative indicators, which prevents it from being a general way to verify the conclusions in this paper.**
>
> In the latest version, we have removed all subjective statements and made the discussion factual using newly-proposed quantitative indicators and statistical correlations for the defined properties.
> We have added more interesting findings as stated above thanks to the extra page available page (we have also moved one of the original subsections to the appendix to save more space for analysis).

---

### Official Review · AnonReviewer2 · 2020-10-29
**interesting study of PEs**

**Rating:** 8
**Confidence:** 4

**Review:**

Updates after author responses and revisions:

I was positive about this paper previously and am glad to see that the authors have done a great deal to try to respond to our concerns and strengthen the paper. I am more positive about the paper now and have increased my score to an 8. I think this paper is going to be useful for the community and I know I will reference it later and direct others to it who are interested in learning more about position embeddings in transformers (whether or not it actually gets published).

--------------

This paper studies position embeddings (PEs) in transformers, suggesting a few reasonable formal properties of PEs and determining whether these properties are captured by various choices for defining PEs. The paper considers both absolute and relative PEs, and both fully-learnable and sinusoidal. A new variation is to learn frequencies in the sinusoidal PEs. Experiments are conducted by training BERT with various choices for PEs and evaluating on GLUE tasks and SQuAD. Fully-learnable absolute PEs, the default in BERT, works quite well overall, but adding learnable sinusoidal relative PEs as well can improve performance on average. There are also visualizations of PE dot products for the various methods, showing that learning PEs in the context of BERT training can yield PEs that satisfy the formal properties (for the most part) laid out by the authors.

I really enjoyed reading this paper. PEs in BERT have been the subject of a lot of informal discussion over the past couple of years, but I don't think I've read a paper that studies the topic with such breadth and depth. With some doable improvements and clarifications, I think the paper can become an excellent resource for others interested in representing position and distance in transformer-like models.

I like the idea of learned sinusoidal embeddings and think that idea can be potentially useful for other researchers. The paper notes that with the fixed sinusoidal embeddings, distances beyond about 50 are not distinguished, but with learning they can be. The experiments show that learning frequencies in sinusoidal PEs works better than using fixed sinusoidal PEs. I'm curious what the learned frequencies look like and how similar they are to the original frequencies chosen by the transformer authors.

Some other thoughts I had while reading:

At least for language tasks with a given window, could we just re-use the learned frequencies and use fixed sinusoidal PEs in the future with those same learned frequencies? I guess a similar choice can be made with fully learned PEs. Do we really need to learn PEs from scratch every time? What aspects of the data or task would influence this? Maybe given their experience, the authors could hand-design useful general-purpose PEs?

Based on the discussion in Sec. 5.3, maybe a good recommendation for BERT-like models would be to use a single learnable PE only for [CLS] and something else (e.g., learned sinusoidal APEs or RPEs) for other positions? This seems easy enough to do in practice and may bring the best of both worlds.


Below are some questions and points of confusion I had:

In the fully-learnable APE experiments described in Sec 4.2, for the first 10 epochs, were the unseen PEs randomly initialized and finetuned in the downstream SQuAD experiments?

I'm confused as to why some of the visualizations in Fig 3 show white bands along the diagonal (d, f, and g). I would have expected all to have dark bands along the diagonal (as in b, c, d, and h).

I was super confused by the identical word probing parts of the paper. Sec. 3.1 includes a sentence beginning with "This shows that the selection of words". The sentence is awkward and confusing to me. I don't know what the "This" is referring to. It seems like there should have been a result reported there for the "This" to refer to. The following sentence, starting with "Namely", is also confusing to me. The description of identical word probing then points to Sec 5.2, and the Sec 5.2 section title is "IDENTICAL WORD PROBING", but it seems to be focused on describing Fig 3, which has as its caption "Dot products between relative position vectors". Where are the identical word probing results actually reported?

In the "Pre-training" paragraph of Section 4, it is mentioned that a pretrained BERT checkpoint is used, but also that other BERT models with other position embeddings were trained. How was the pretrained model actually used? Was it used to initialize the other models or were all models trained from scratch?

Minor things:

In the first equation in Sec 3.2, there are a couple of instances of K/2 which I think should be D/2 instead. Also, the vectors start with \omega_1 but the summation at the end of the equation starts with \omega_0.

I'm confused by the bolding in Table 2. The best number in each column is not always in boldface (QNLI, WNLI, etc.). Also, sometimes when there is a tie for the best result, multiple numbers are in bold, while other times only one result is in bold.


Typos:

p. 1: "constrains" --> "constraints"

footnote 4: "seeing" --> "See"

Sec. 4.1: "outperform notably" --> "notably outperform"

Sec. 5.3: "PEs o" --> "PEs to"

---

> ### Author Response · Authors · 2020-11-23
> **Reply to Reviewer2's thoughts**
>
>
> We thank you for the helpful comments and suggestions.
> We provide our responses as follows.
>
> **I'm curious what the learned frequencies look like and how similar they are to the original frequencies chosen by the transformer authors**
>
>
> In appendix A3, Figure 4(a)  shows the learned frequencies, and Figure 4(b) shows how dot product between two n-distance position vectors learned from pre-trained models and fine-tuned tasks. The difference between the original one is not notable,  probably due to the fact that we initialize it using the frequencies of original transformers Vaswani et al.\ ( $\omega_i = ({1}/{10000})^{2i/D}$).
>
> It is worth noting that Parascandolo et al.\ observed that  there are infinitely many and shallow local minima during training with  sine/cosine functions; therefore, having a good  initialization (like  $\omega_i = ({1}/{10000})^{2i/D}$) may be important. Anyway, we will release the result which trains from scratch (random initialization for frequencies and training BERT from scratch) in the next version and keep you updated, since it takes a few weeks.
>
> [1] Vaswani, Ashish, Noam Shazeer, Niki Parmar, Jakob Uszkoreit, Llion Jones, Aidan N. Gomez, Łukasz Kaiser, and Illia Polosukhin. "Attention is all you need." In Advances in neural information processing systems, pp. 5998-6008. 2017.
>
> [2] Parascandolo, Giambattista, Heikki Huttunen, and Tuomas Virtanen. "Taming the waves: sine as activation function in deep neural networks." (2016).
>
>
> **Thought1 : Do we really need to learn PEs from scratch every time?**
>
>
> Generally, we need to learn PEs from scratch for individual tasks since it could fit the tasks better. Wang & Chen reuses these PEs as initialization in various tasks and see  difference in performance.
>
> Wang, Yu-An, and Yun-Nung Chen. "What Do Position Embeddings Learn? An Empirical Study of Pre-Trained Language Model Positional Encoding." arXiv preprint arXiv:2010.04903 (2020). Accepted by EMNLP, publicly available after the ICLR deadline.
>
>
> **Thought2: What aspects of the data or task would influence this? **
> As concluded in the revised paper,   both classification and span prediction benefit from the translation invariance and local monotonicity properties of PEs, while symmetry slightly decreases their performance because of the inability to perceive directions. See. Tab. 5 for more details.
>
> **Thought3: Maybe given their experience, the authors could hand-design useful general-purpose PEs?**
>
>
> For APEs, ideally,  we suggest that it should: (1) untie the  [CLS] with other positions like Ke et al. 2020; (2) For any position embeddings $P \in  \mathbb{R}^{L \times D}$, the outer product between itself should result in a Toeplitz matrix (a matrix in which each descending diagonal from left to right is constant):
>
> $P P^T = \begin{bmatrix} r_0 & r_1 	& \cdots 	& r_{L-1}   	\\\\  r'_{-1} 	& r_0 	& \cdots 	& r_{L-2}  \\\\ \cdots 	& \cdots 	& \cdots 	& \cdots    \\\\     r'_{-L+1}  & r'_{-L+2} 	& \cdots 	& r_0	\\\\      \end{bmatrix}$ ( The matrix has some edit error for last row, see following matrix by replacing the last  row with the one in this line)
>
> $P P^T = \begin{bmatrix} r_0 & r_1 	& \cdots  	& r_{L-1} \\\\ r'_{-1} 	& r_0 	& \cdots	   &  r_{L-2}   \\\\  \cdots 	& \cdots 	& \cdots 	& \cdots    \\\\   \cdots & \cdots   &   \cdots      & r_0  \\\\    \end{bmatrix}$
>
> such that
> $r_0 >  r_{-1} > \cdots > r_{-L+1} $ and
> $r_0 >  r_{-1}' > \cdots > r_{-L+1}' $
>
> Comparing to APEs, RPEs are more preferable since they satisfy translation invariance during parameterizations. The other two properties are somehow too complicated to analyze.
>
>
> **Thought4: Based on the discussion in Sec. 5.3, maybe a good recommendation for BERT-like models would be to use a single learnable PE only for [CLS] and something else (e.g., learned sinusoidal APEs or RPEs) for other positions? This seems easy enough to do in practice and may bring the best of both worlds.**
>
> We totally agree. A very recent paper (Ke et al. 2020) combines CLS with absolute position embeddings; it would be nice to try this with RPEs. We have also discussed it in Appendix H of our paper (and we believe that our paper provides additional motivation for (Ke et al. 2020)).
>
> Ke, Guolin, Di He, and Tie-Yan Liu. "Rethinking the Positional Encoding in Language Pre-training." arXiv preprint arXiv:2006.15595 (2020).

---

> ### Author Response · Authors · 2020-11-23
> **Reply to Reviewer2's questions and  confusion**
>
>
> **Q1:  In the fully-learnable APE experiments described in Sec 4.2, for the first 10 epochs, were the unseen PEs randomly initialized and fine-tuned in the downstream SQuAD experiments?**
>
> Yes, the unseen PEs are randomly initialized and fine-tuned in the downstream SQuAD experiments. We have clarified this in the latest version.
>
> **Q2: I'm confused as to why some of the visualizations in Fig 3 show white bands along the diagonal (d, f, and g). I would have expected all to have dark bands along the diagonal (as in b, c, d, and h).**
>
> In the latest version, we have investigated this "white band" phenomenon using identical word probing tests with (an average of) larger-scaled and more diverse words (see * newly-added Appendix J*). The cause of the effect is that some words generally do not attend themselves, as also reported in (Clark et al., 2019).
>
> Clark, Kevin, Urvashi Khandelwal, Omer Levy, and Christopher D. Manning. "What does bert look at? an analysis of BERT's attention." arXiv preprint arXiv:1906.04341 (2019).
>
> One interesting point is that the white band effect only involves learnable RPEs: the more trainable parameters in the RPE, the more notable the effect becomes. Hence, the effect is stronger in fully-learnable RPEs than in learnable sinusoidal RPEs, which in turn exhibit the effect more strongly than fixed sinusoidal RPEs. We leave the further investigation of this for future work.
>
> **Q3: I was super confused by the identical word probing parts of the paper. Sec. 3.1 includes a sentence beginning with "This shows that the selection of words". The sentence is awkward and confusing to me.  ...**
>
> We apologize for it,  we have improved the writing in the latest version, and have carefully proofread the original text.
>
> The caption to describe Fig.3  was wrong; Fig.3  was supposed to concern identical word probing of A. And we rewrote "this shows ...".
>
> **Q4: In the "Pre-training" paragraph of Section 4, it is mentioned that a pre-trained BERT checkpoint is used, but also that other BERT models with other position embeddings were trained. How was the pre-trained model actually used? Was it used to initialize the other models or were all models trained from scratch?**
>
> In all BERT models with different PE variants, we reuse most of the trained parameters (except for PE components) of a BERT base checkpoint as initialization, while PEs were re-designed. In the BERT-base setting, they are not trained from scratch, which takes a longer time to train. In the BERT-medium setting, we instead tried to train from scratch as this could potentially be more efficient (training on the BERT medium setting is in general much faster).
>
> We have clarified the above in the revised version.

---

### Official Review · AnonReviewer1 · 2020-10-29

**Rating:** 7
**Confidence:** 4

**Review:**

The paper presents a systematic analysis of approaches used to encode position information in transformers and in particular BERT-based models. The paper investigates absolute and relative position embedding strategies that use either fixed/learnable sinusoidal or fully learnable position embeddings. These embeddings are characterized based on different properties that are either inherent from their formulation or observed empirically such as monotonicity, translation invariance, and symmetry. Interestingly these properties appear to emerge naturally when having learnable parameters in APEs and RPEs.

Different PE strategies are empirically validated by pre-training BERT with different PEs (including a combination of absolute and relative position representations) and fine-tuning on GLUE and SQuAD (1.1 and 2.0). Visualizations of the dot products between position vectors for different PE strategies are presented as well that demonstrate the monotonicity (or local monotonicity), symmetry, and translation invariance.

Overall, the paper is well written, motivated, and systematically studies an important design decision in transformers. The overall methodology is sound and should prove useful to the community when studying position embeddings in transformers.

Strengths

The paper is well written, well-motivated, and is systematic in its claims, experiments, and methodology.
It takes a good step forward in characterizing desirable properties of position embeddings and studying if they emerge directly from their parameterization or via training.
It studies a variety of position embedding strategies and their conjunction and experiments with fairly realistic models and benchmarks.

Weaknesses

While the analyses are well carried out, they are still specific to BERT-like masked language modeling training and it isn’t clear if PEs behave similarly in encoder-decoder like models for summarization/translation or decoder only language models.
It does not control for the choice of (pre)training objective, for example, if trained from scratch, purely as a retrieval model or on supervised text classification, would different learned PE patterns emerge?

It provides certain properties one can hope PEs to exhibit but doesn’t talk too much about circumstances under which these would be good inductive biases to have.

Questions & Comments:

Under what circumstances do properties like monotonicity in learnable PEs arise? Would PEs for an autoregressive variant also display similar properties? (would be interesting to see what happens with say a transformer LM trained from scratch on wikitext-103)

It isn’t quite clear to me at first glance why the preservation of the order of distances is necessary for position embeddings when used with models that build nonlinear functions of these embeddings.

Would absolute position embeddings that are randomly initialized or all initialized orthogonal to one another (something like 1-hot vectors of dimension L) *and fixed*  work? This would be interesting to see (especially the latter) because it does not satisfy monotonicity or translation invariance and violates the desiderata in eq (1)

Is Figure 7 based on BERT fine-tuned for NER on the CONLL/Ontonotes dataset?

---

> ### Author Response · Authors · 2020-11-23
> **In the revision, we quantitatively concluded how these properties benefit/harm downstream tasks**
>
> We thank you for the helpful comments and suggestions.
> We provide our responses as follows.
>
>
>
>  **Weaknesses1: the analyses are specific to BERT-like masked language modeling, it isn’t clear if PEs behave similarly in encoder-decoder like models for summarization/translation or decoder only language models**
>
>
> In the latest version, we conduct the probing tests for well-trained GPT2 and English-to-French machine translation models. Please see Appendix I.
>
> We found that there are some differences:
> 1. Monotonicity: compared to BERT and the machine translation, GPT2  satisfies local monotonicity  better than other models, showing capturing distance between neighboring tokens matters in language model.
>
> 2.Translation invariance:   GPT2 satisfies Translation invariance poorer than other models, since tokens in it also additionally attend to a few beginning tokens no matter how far the attended tokens are.
>
> 3 Direction Awareness: GPT2 shows the biggest symmetrical discrepancy, since GPT2, which aims to predict the next word, adopts an attention mask of succeeding tokens to avoid information leakage. Plus, MT Encoder slightly attend more to the succeeding tokens while BERT attend more on the preceding tokens
>
>
>
> **Weaknesses2: It provides certain properties one can hope PEs to exhibit but doesn’t talk too much about circumstances under which these would be good inductive biases to have.**
>
> In the latest version, we have adopted multiple quantitative indicators to measure how well BERT models with individual PEs satisfy the three properties. In addition, we compute correlations between said indicators and performance: the result of this investigation is that both the text classification tasks (in GLUE) and span prediction tasks (in SQuAD V1.0 and V2.0) can benefit from local monotonicity and translation invariance, but that symmetry results in performance deterioration, possibly due to the inability of symmetric functions to properly reflect the transition from  query vector to key vector when calculating attention. We believe that this conclusion may also hold for other tasks, but have not performed an experimental evaluation of this. The next reply also helps answer the question regarding circumstances.
>
>
>
> **Q1: Under what circumstances do properties like monotonicity in learnable PEs arise? Would PEs for an autoregressive variant also display similar properties? (would be interesting to see what happens with say a transformer LM trained from scratch on wikitext-103)**
>
> (1) Monotonicity (especially in  small offsets) should generally arise if tasks need to capture word order (most tasks should be). Removing monotonicity suggests that the models become similar to bag-of-words models that do not consider word order.
>
> (2) Translation invariance probably arises due to the input data processing in BERT. Absolute positions of words in pre-trained language models (such as BERT) are arbitrarily replaceable. See the discussion on translation invariance in Sec 6.
>
> (3)  Symmetry should in general be avoided because it degrades performance. It is likely that it cannot properly express directionality when calculating attentions between query vectors and key vectors.
>
> **Q2: It isn’t quite clear to me at first glance why the preservation of the order of distances is necessary for position embeddings when used with models that build nonlinear functions of these embeddings.**
>
> It is quite difficult to make this evident theoretically, unless we assume more about the particular non-linear functions. We empirically find that the learned PEs preserve the order of distance. We have added a brief comment to that effect in the paper.
>
>
> **Q3: Would absolute position embeddings that are randomly initialized or all initialized orthogonal to one another (something like 1-hot vectors of dimension L) and fixed work? This would be interesting to see (especially the latter) because it does not satisfy monotonicity or translation invariance and violates the desiderata in eq (1)**
>
> For random initialization, see Fig. 8 in App. G, showing the dot products between any two position vectors in a BERT medium training from scratch. We could see a  gradual evolution of dot products between position vectors. At the onset, it indeed does not satisfy monotonicity or translation invariance, but with an increasing number of training steps, we observe clearly local monotonicity and translation invariance.
>
> **Q4: Is Figure 7 based on BERT fine-tuned for NER on the CONLL/Ontonotes dataset?**
> No, these figures are for BERT pre-trained models without fine-tuning.  We have clarified this in the latest version.

---

### Official Review · AnonReviewer4 · 2020-10-29
**Interesting Take on Comparing Position Embeddings**

**Rating:** 6
**Confidence:** 4

**Review:**

This paper proposes a formal framework to compare position embeddings (PEs) and presents an empirical study comparing variants of absolute position embeddings (APEs) and relative position embeddings (RPEs) on three properties: 1) monotonicity, 2) translation invariance, and 3) symmetry and evaluates their performance on classification (GLUE) and span prediction tasks (SQUAD). The authors also report results on learnable sinusoidal APEs and learnable sinusoidal RPEs, PE variants which had not been previously proposed.

The first three properties seem well-motivated (monotonicity and translation invariance), but it is not obvious that symmetry should be a property of an ideal PE, or at least the paper is not convincing on this front. In a sentence (ABCD), doesn’t the word A typically have a different relationship to B than B does to A?

The identical word probing test was a clever way to disentangle the impact of the word from that of the PEs.

While it does seem valuable to more rigorously compare PEs as they are critical components of SOTA language models, the experimental results were not particularly convincing (e.g. although it’s a very appealing story, it didn’t seem so clear from the tables that APEs did better at classification and that RPEs did better at span prediction.)

The writing quality was borderline and there were a number of small errors:
- fully-learnable APEs nearly meet all properties even under no constrains” -> “constraints”
- Under the equation at the beginning of Section 3.1, “word-word correspondence” is repeated four times, which I am sure was not the intention.
- nit: “since relative distance with the same offset will be embedded as a same embedding.” -> “the same embedding”
- “compared to far-way” -> “faraway”
- “attends more on forwarding tokens than backward tokens” -> “forward tokens”?
- nit: “In Transformer, where attention calculation does not…” -> “where the attention calculation”
- “allows PEs o better perceive word order” -> “to”

---

> ### Author Response · Authors · 2020-11-23
> **Reply to AnonReviewer4**
>
> We thank you for the helpful comments and suggestions. We provide our responses as follows.
>
> **Q1: assume  symmetry by default**
>
> In the latest version, we now clearly state that the three properties are not gold standards for any position embedding, but that the properties are worthy and interesting to examine in light of the previous literature on position embeddings.  Finally, the empirical results did evidence that directions matter, that is, `word A does have a different relationship to B than B does to A', as you suggested.
>
> There are two reasons to test   symmetry for PEs:
>
>  (**1**)  Denote by $\phi(\cdot,\cdot)$ some function that calculates closeness/proximity between embedded positions; in the literature, $\phi(\cdot,\cdot)$ is typically a mapping $f: \mathbb{R}^D,  \mathbb{R}^D \rightarrow \mathbb{R}^+  $. This is also consistent with attention calculation, for example $\textrm{softmax} \left((w_i+p_i) W^{Q,1} ((w_j+p_j) W^{K,1} )^T\right)$ (note that this involves a position-position   correspondence $p_i W^{Q,1} (W^{K,1})^T  p_j ^T$ ) also returns a positive real output.
> As laid out in the paper, attention values of both forward and backward attending need to be mapped to positive reals, but unless more is known about $\phi(\cdot,\cdot)$, the function itself does not necessarily yield any information about directionality.
>  (**2**) a typical and well-defined way to choose a proximity function $\phi(\cdot,\cdot)$ is to use an inner product (for example, the dot product $p_i  p_j ^T$ ), and (real-valued) inner products by definition satisfy symmetry.
>
>
>
> **Q2: The experimental results were not particularly convincing. Conclusion about APEs did better at classification and that RPEs did better at span prediction.**
>
>
> The conclusion we made was that the **fully-learnable APE** (instead of general **APEs**) performed better at classification. BERT with the **fully-learnable APE** is the baseline for all PE variants because (1) it is implemented in the original BERT, and (2) it is simple and efficient. In the latest version, we have rewritten the related content to make the above more clear.
>
> We made the conclusion above because:
>
> In classification tasks (GLUE), No PE variants significantly outperform the * fully-learnable APE*.
> (1) In all BERT models using a single type of PEs (without considering combinations between APEs and RPEs), all PE variants (including the fixed sin. APE, learnable sin. APE, fully-learnable RPE, fixed sin. RPE, and learnable sin. RPE)  perform worse than the fully-learnable APE, as shown in first seven rows of Table 3.
> (2) Although an APE-RPE mixed variant (*learnable sin. APE + fully-learnable RPE*) performs better than fully-learnable  APE (82.5 vs 82.2), but this  improvement is not statistically significant.
>
> In span prediction, without considering APE-RPE mixed variants, all RPE variants significantly outperform APE variants.
>
>
>
>  **Q3: The writing quality was borderline and there were a number of small errors**
>
> The paper has now been carefully proofread.

---

### Author Response · Authors · 2020-11-24
**The end of disucssion phase approaching**

Dear Reviewers and Area Chair,


Could you please go over our responses and revision to have more possible interactions before this Tuesday (24th Nov.)?
Based on the first-round reviews, we have provided the suggested experiments, quantitative analysis, detailed explanations, and some interesting discussion.

Thanks for your time reviewing the paper, and your comments have substantially improved the quality of this paper.


In the revision,

(1) We have added quantitative measurement for the three properties in Sec. 4.1.2 (Table 2) and Appendix B;

(2) We have conducted an analysis of the correlations between properties and performance in individual tasks, in Sec. 4.3 (Table 4).

(3) We have investigated the difference between learned frequencies and predefined ones (Vaswani et al. 2017) in Appendix A.3.

(4) We have shown how models are insensitive to long-distance offsets in Appendix C.

(5) We have investigated how PEs evolve with an increasing number of training steps in Appendix F, in particular, PEs gradually increasingly satisfy some of the three main properties we examine in the paper.

(6)The white band effect indicates words do not attend themselves in learnable RPEs, as previously stated by Clark et al. 2019.

(7)Comapring the position related attention patterns with GPT and a machine translation model in App. I

Also, we carefully proofread the whole paper: fixed language iusses and removed all subjective statements to make the discussion factual.

Best,

The authors

---

### Decision · Program_Chairs · 2021-01-07
**Final Decision**

**Decision:**

Accept (Poster)

**Comment:**

With the advent of non-recurrent sequence-processing models, it has become costumary to augment input tokens with positional embeddings providing implicit positional information. Despite their potentially crucial role in modern architectures, such positional embeddings are rarely addressed in analytical studies. The current paper provides a systematic investigation of positional embeddings, characterized in terms of properties such as monotonicity, translation invariance, and symmetry. These properties are studies for different positional embeddings using language models fine-tuned on two separated benchmarks, with an emphasis on visual analysis.

The authors provided an impressive rebuttal, adding many of the experiments required by the reviewers. The latter are still somewhat split about the paper. I lean towards the positive side. I find that some of the criticism, while valid, is not really granting a rejection, especially after the authors' clarifications. In particular, one reviewer assumed that the authors claim that symmetry should be a property of an ideal positional embedding, whereas the authors are rather studying whether it is an important property of them, in light of the previous literature. Some claims about the results being "interesting" or "surprising" enough might depend on what the reader is looking for in the paper. I think that many readers in the "black box NLP" community will find the methods and analyses presented in this paper interesting and useful.